# *MECP2* mutations affect ciliogenesis: a novel perspective for Rett syndrome and related disorders

Angelisa Frasca[1,†] , Eleonora Spiombi[1,†], Michela Palmieri[2], Elena Albizzati[1], Maria Maddalena Valente[3], Anna Bergo[3], Barbara Leva[3], Charlotte Kilstrup-Nielsen[3], Federico Bianchi[4], Valerio Di Carlo[1], Ferdinando Di Cunto[4,5] & Nicoletta Landsberger[1,2,*]

## Abstract

Mutations in *MECP2* cause several neurological disorders of which Rett syndrome (RTT) represents the best-defined condition. Although mainly working as a transcriptional repressor, MeCP2 is a multifunctional protein revealing several activities, the involvement of which in RTT remains obscure. Besides being mainly localized in the nucleus, MeCP2 associates with the centrosome, an organelle from which primary cilia originate. Primary cilia function as "sensory antennae" protruding from most cells, and a link between primary cilia and mental illness has recently been reported. We herein demonstrate that MeCP2 deficiency affects ciliogenesis in cultured cells, including neurons and RTT fibroblasts, and in the mouse brain. Consequently, the cilium-related Sonic Hedgehog pathway, which is essential for brain development and functioning, is impaired. Microtubule instability participates in these phenotypes that can be rescued by HDAC6 inhibition together with the recovery of RTT-related neuronal defects. Our data indicate defects of primary cilium as a novel pathogenic mechanism that by contributing to the clinical features of RTT might impact on proper cerebellum/brain development and functioning, thus providing a novel therapeutic target.

**Keywords** MeCP2; primary cilium; Rett syndrome; sonic hedgehog; tubacin treatment

**Subject Categories** Genetics, Gene Therapy & Genetic Disease; Neuroscience

## Introduction

The Methyl-CpG-binding Protein 2 (*MECP2*) gene, localized on the X chromosome, codes for a multifunctional epigenetic regulator with a prominent role in brain. Accordingly, *MECP2* mutations are linked to several neurological conditions characterized by cognitive impairment and intellectual disability (Ezeonwuka & Rastegar, 2014). In particular, loss-of-function mutations are mainly associated with Rett (RTT) syndrome, a severe neurodevelopmental disease that principally affects females (Amir *et al*, 1999; Chahrour & Zoghbi, 2007). Less frequently, *MECP2* mutations cause autism, schizophrenia, mental retardation, Angelman-like syndrome in both genders and neonatal encephalopathy in males (Ezeonwuka & Rastegar, 2014). In parallel, a non-physiological increase in MeCP2 expression is responsible for the recently identified *MECP2* duplication syndrome, mainly affecting males (Ramocki *et al*, 2010). In accordance with its ubiquitous expression, *MECP2* has also been linked to non-neurological diseases, such as lupus erythematosus, rheumatoid arthritis and cancer (Ezeonwuka & Rastegar, 2014).

Originally isolated as the first protein able to specifically bind methylated cytosines, MeCP2 is generally described as an epigenetic transcriptional regulator that represses transcription of methylated DNA. This repressive activity is mainly mediated by the ability of MeCP2 to recruit corepressor complexes able to modify chromatin structure (Clouaire & Stancheva, 2008). In addition to its proposed role in gene silencing and chromatin architecture, several other functions have more recently been ascribed to MeCP2. Indeed, nowadays MeCP2 appears as a multifunctional protein that manifests different activities depending on its partners and post-translational modifications (Young *et al*, 2005; Chahrour *et al*, 2008; Ghosh *et al*, 2010; Skene *et al*, 2010; Bedogni *et al*, 2014; Bellini *et al*, 2014).

Accordingly, we have recently demonstrated that MeCP2 is functionally associated with centrosomes and, similarly to many

1  Department of Medical Biotechnology and Translational Medicine, University of Milan, Milan, Italy
2  Neuroscience Division, IRCCS San Raffaele Scientific Institute, Milan, Italy
3  Department of Biotechnology and Life Sciences, Centre of Neuroscience, University of Insubria, Busto Arsizio, Italy
4  Neuroscience Institute Cavalieri Ottolenghi, Orbassano, Italy
5  Department of Neuroscience, University of Torino, Torino, Italy
   *Corresponding author. Tel. +39 0250330462; E-mail: nicoletta.landsberger@unimi.it
   †These authors contributed equally to this work as first authors

centrosomal proteins, its deficiency causes aberrant spindle geometry, defects in cell proliferation, microtubule nucleation and prolonged mitosis. This novel function reconciled previous data regarding a role of MeCP2 in cell growth, cytoskeleton stability, and axonal transport (Bergo *et al*, 2015). While in cycling cells, the centrosome organizes the interphase microtubule network and the mitotic spindle, in most quiescent or post-mitotic non-cycling cells, it participates in the nucleation of primary cilia (Stearns, 2001). We thus decided to investigate whether MeCP2 defects could play a role also in primary cilium formation and functioning.

Primary cilium is a non-motile microtubule-based organelle that grows from a centrosome-derived structure, termed the basal body, and protrudes from the cell surface of almost every quiescent or differentiated mammalian cell, including neurons (Guemez-Gamboa *et al*, 2014). It consists of an axoneme, formed by a radial array of nine doublet microtubules of polymerized α/β-tubulin heterodimers, and the process of intraflagellar transport (IFT) is responsible for building and maintaining cilium structure and function (van Reeuwijk *et al*, 2011; Pala *et al*, 2017; Youn & Han, 2018). Primary cilium is a sensory organelle that receives and transduces many extracellular signals, thereby influencing a wide variety of cellular processes, such as proliferation, differentiation, migration, and apical–basal polarity in cortical development and neuronal growth (van Reeuwijk *et al*, 2011; Pala *et al*, 2017; Youn & Han, 2018; Park *et al*, 2019). In accordance with its main function, several receptors for signaling cascades, such as Sonic Hedgehog (Shh), PDGF, mTOR, Wnt, Notch, planar cell polarity (PCP) and Hippo, accumulate in the membrane of primary cilium (Goetz & Anderson, 2010; Pala *et al*, 2017; Wheway *et al*, 2018). The importance of this organelle in normal physiology is exemplified by a growing list of genetic disorders, known as "ciliopathies", caused by dysfunctional ciliary assembly, anchoring and/or signaling, characterized by a high variability in clinical presentation (Novarino *et al*, 2011; Ware *et al*, 2011; Lee & Gleeson, 2014; Valente *et al*, 2014; Reiter & Leroux, 2017). Interestingly, among the variety of clinical features, some symptoms such as developmental delay, cognitive disabilities, ataxia, tendency for obesity, bone loss and defects in respiration rhythms are also manifested by RTT patients (Kyle *et al*, 2018). Furthermore, a recent study has indicated that among 41 genes connected to diverse neuropsychiatric disorders, 23 are involved in primary cilium assembly and disassembly (Marley & von Zastrow, 2012).

Cilium assembly/disassembly depends on a great number of different factors that have to be optimally tuned in order to generate cilia of the right size and length, thereby ensuring proper functioning (Keeling *et al*, 2016). Among these factors, acetylation of lysine (K) 40 of α-tubulin is pivotal for maintaining primary cilium; indeed, its HDAC6-dependent deacetylation leads to the disassembly of the organelle (Forcioli-Conti *et al*, 2016). Importantly, *MECP2*-deficient cells, including fibroblasts from RTT patients, are characterized by the upregulation of HDAC6 and, consequently, reduced acetylation of polymerized α-tubulin and microtubule instability (Delépine *et al*, 2013, 2016; Gold *et al*, 2015).

Considering all above, for the first time, we have investigated and identified primary cilium dysfunctions *in vitro* in all tested cells, including fibroblasts from RTT patients, and *in vivo* in *Mecp2* null and heterozygous brains, demonstrating a causal connection between MeCP2 expression and ciliogenesis. Importantly, these defects reflect, both *in vitro* and *in vivo*, a functional impairment of the ciliary-related Shh signaling pathway. Stabilization of α-tubulin, through a selective inhibition of HDAC6, can revert the observed morphological and functional ciliary alterations, in concomitance with a recovery of RTT-related phenotypes in *Mecp2* null neurons.

# Results

### Primary cilium formation is facilitated by MeCP2

As mentioned above, we have recently demonstrated a molecular and functional association between MeCP2 and the centrosome, the cellular organelle that templates the assembly of primary cilium (Bergo *et al*, 2015). Almost every quiescent or post-mitotic mammalian cell presents a primary cilium; we thus investigated whether MeCP2 deficiency affects the presence and/or length of the primary cilium on the cell surface.

First, we compared wild-type (WT) and *Mecp2* null mouse quiescent embryonic fibroblasts (MEFs). Ciliated cells were detected by immunofluorescence staining for acetylated α-tubulin and γ-tubulin, two microtubule proteins that are enriched, respectively, in the axoneme and the basal body of the cilium, where they are critical for maintaining its structure (Fig 1A). As shown in Fig 1B, the percentage of ciliated cells was significantly decreased by 38% compared to WT cells (**$P < 0.01$, Mann–Whitney test). Since almost 53% of *Mecp2* null cells showed a primary cilium, we measured its length. Nascent or extended acetylated α-tubulin emerging from the γ-tubulin-positive centriole was measured, revealing that *Mecp2* null MEFs show a statistically significant reduction in cilium length (***$P < 0.001$, Student's *t*-test; Fig 1C). Similar results were obtained analyzing MEFs from a $Mecp2^{Y120D/y}$ knock-in mouse mimicking a pathogenic human mutation (*$P < 0.05$, Mann–Whitney test; Fig EV1; Gandaglia *et al*, 2018). To support these data, we repeated the same analyses in hTERT-RPE-1 (immortalized retinal pigment epithelial) cells interfered with a specific MeCP2-siRNA, reducing MeCP2 expression to less than half, as revealed by Western blot (Fig 1E). Accordingly, immunofluorescence analysis proved that almost 40% of cells show a positive MeCP2 signal (***$P < 0.001$, Student's *t*-test; Fig 1F). Primary cilia were stained and analyzed (Fig 1D). Quantitative analyses indicated that the percentage of ciliated cells was significantly reduced compared to control (***$P < 0.001$, Mann–Whitney test; Fig 1G) and, when present, cilium length was significantly decreased in silenced cells (***$P < 0.001$, Student's *t*-test; Fig 1H). Since Rett syndrome is a neurological disorder affecting both neurons and astrocytes, we also analyzed ciliogenesis in these cells. Primary neurons derived from WT or *Mecp2* null embryonic cortices were fixed at different maturation stages and cilium was detected by immunostaining for adenylate cyclase type 3 (AC3), a protein expressed in the primary cilium of neurons (Bishop *et al*, 2007; Fig 1I). Indeed, it is well recognized that the antibody against acetylated α-tubulin, which is commonly used to mark the primary cilium in several non-neuronal cell types, in neurons causes a diffuse staining throughout the cell body and neurites, thus preventing its use as valid cilium marker (Berbari *et al*, 2013). The staining was conducted at DIV3, DIV7 and DIV14. Results revealed an impairment in the percentage of ciliated cells and in cilium length

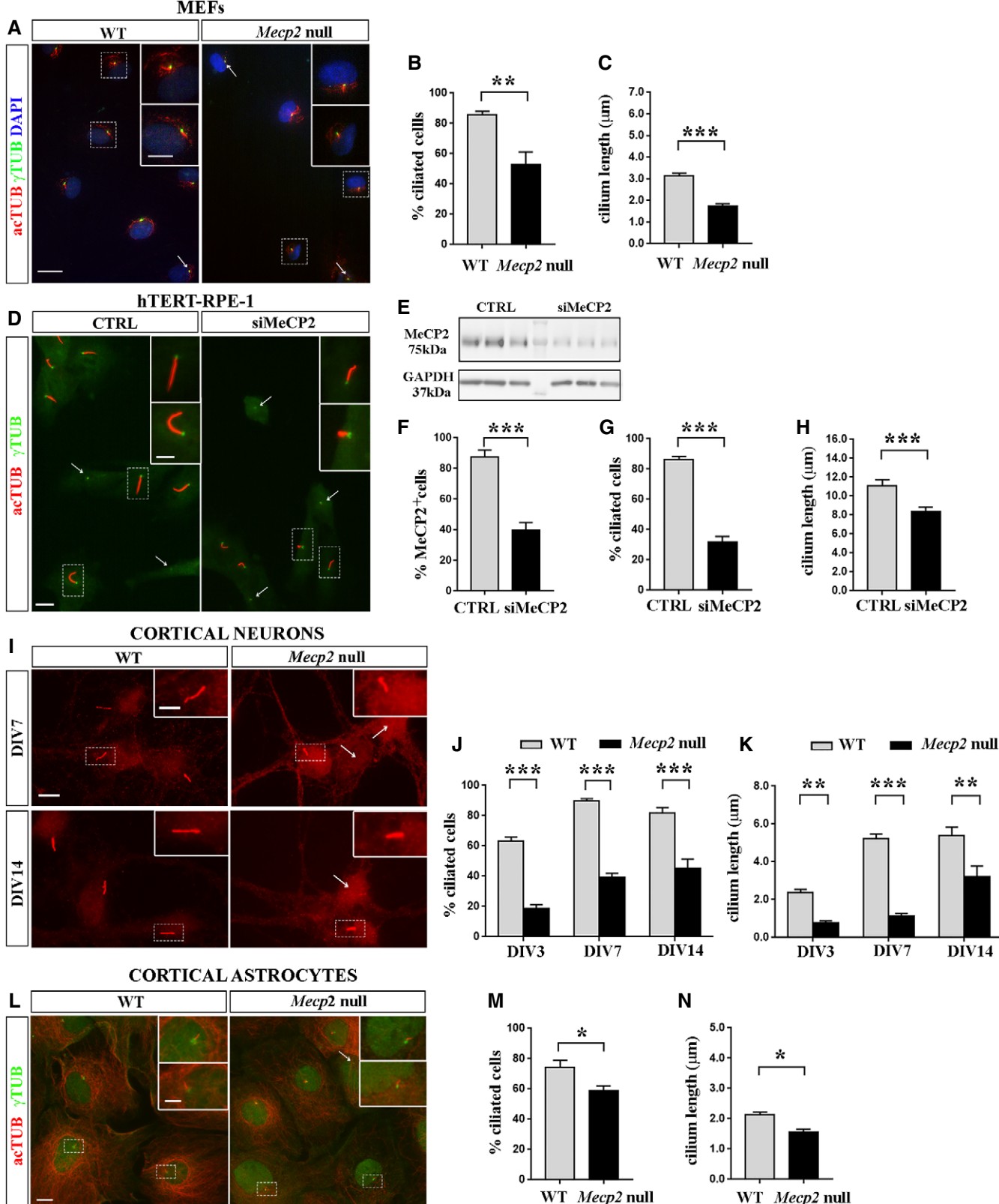

**Figure 1.**

---

**Figure 1.  MeCP2 deficiency affects primary cilium formation.**

A    Representative immunofluorescence of primary cilium in WT or *Mecp2* null MEF cells. Cells were starved for 48 h before staining with anti-γ-tubulin (green) and anti-acetylated α-tubulin (red) antibodies. Merge of all channels and DAPI staining (blue) is depicted for both genotypes. Scale bar 20 μm, and 10 μm in the enlarged image. Arrows indicate not ciliated cells.

B, C    Cilia were counted and analyzed with ImageJ to quantify the percentage of ciliated cells ($n = 134$ WT and $n = 140$ *Mecp2* null cells) (B) and the cilium length ($n = 104$ WT and $n = 43$ *Mecp2* null cells) (C). The graphs show the average of three independent experiments (mean ± SE, **$P < 0.01$, Mann–Whitney test; ***$P < 0.001$, Student's *t*-test).

D    Representative immunofluorescence of primary cilium in hTERT-RPE-1 cells silenced with *MECP2* or control siRNAs. Cells were starved for 48 h before staining with anti-γ-tubulin (green) and anti-acetylated α-tubulin (red) antibodies. Scale bar 10 μm, and 5 μm in the enlarged image. Arrows indicate not ciliated cells.

E    Representative Western blotting showing the MeCP2 expression in CTRL and silenced hTERT-RPE-1 cells, with respect to GAPDH, used as loading control.

F    The percentage of MeCP2-positive cells was calculated by analyzing $n = 1,000$ CTRL and silenced hTERT-RPE-1 cells (mean ± SE, ***$P < 0.001$, Student's *t*-test).

G, H    Cilia were counted and analyzed with ImageJ to quantify the percentage of ciliated cells ($n = 191$ CTRL and $n = 197$ siMeCP2 cells) (G) and the cilium length ($n = 50$ CTRL and $n = 50$ siMeCP2 cells) (H). The graphs show the mean ± SE of three independent experiments (***$P < 0.001$, Mann–Whitney test (G) or Student's *t*-test (H)).

I    Representative immunofluorescence of primary cilium in WT and *Mecp2* null primary cortical neurons at DIV7 and DIV14. Scale bar = 10 μm, and 5 μm in the enlarged image. Arrows indicate not ciliated cells.

J, K    Neurons were fixed at DIV3, DIV7, and DIV14, and primary cilium was detected by immunostaining with anti-AC3 antibody. Percentage of ciliated cells (J) and cilium length (K) were analyzed. Histograms represent the average of three independent experiments, analyzing neurons derived from 3 different experiments ($n = 156$ WT DIV3; $n = 49$ KO DIV3; $n = 177$ WT DIV7; $n = 82$ KO DIV7; $n = 59$ WT DIV14; and $n = 37$ KO DIV14) (mean ± SE, **$P < 0.01$; ***$P < 0.001$, two-way ANOVA followed by Bonferroni *post hoc* test).

L    Primary cortical astrocytes were fixed at DIV15 and stained with anti-acetylated α-tubulin (red) and anti-γ-tubulin (green) antibodies. Scale bar = 20 μm, and 10 μm in the enlarged image. Arrow indicates not ciliated cell.

M, N    Percentage of ciliated cells (M) and cilium length (N) were analyzed ($n = 26$ WT and $n = 34$ *Mecp2* null astrocytes). Histograms represent the average of three independent experiments (mean ± SE, *$P < 0.05$, Student's *t*-test).

Source data are available online for this figure.

---

(*$P < 0.05$, two-way ANOVA followed by Bonferroni *post hoc* test), indicating a defective ciliogenesis in *Mecp2* null neurons compared to controls, in front of a progressive increase of ciliogenesis in both WT and null neurons along neuronal maturation ($F_{(2, 584)} = 44.06$ for % ciliated cells; $F_{(2, 551)} = 30.68$ for cilium length) (Fig 1J and K). In a similar way, cortical astrocytes were analyzed by immunostaining cilia for acetylated α-tubulin and γ-tubulin (Fig 1L–N) and a significant reduction in both the percentage of ciliated cells and in cilium length was demonstrated (*$P < 0.05$, Student's *t*-test).

To confirm that MeCP2 expression is involved in the observed phenotypes, we performed a rescue experiment on *Mecp2* null primary neurons (Fig 2). WT and *Mecp2* null cortical neurons were infected at DIV0 with lentiviruses expressing GFP or MeCP2iresGFP, and the expression of MeCP2 was verified by Western blot (Fig 2B). At DIV7, neuronal cilium was immunostained for AC3 (Fig 2A) and the percentage of ciliated cells and cilium length were determined only on GFP-positive cells. We confirmed the impairment in primary cilium formation in *Mecp2* null neurons and demonstrated a significant recovery of the phenotype following MeCP2 expression (***$P < 0.001$, two-way ANOVA followed by Bonferroni *post hoc* test) (Fig 2C). Similarly, MeCP2 transduction fully recovered cilium length in *Mecp2* null neurons (***$P < 0.001$; **$P < 0.01$, two-way ANOVA followed by Bonferroni *post hoc* test) (Fig 2D).

### *Mecp2* deficiency affects the cilium-associated Sonic Hedgehog (Shh) signaling pathway

Since our data demonstrate that MeCP2 deficiency is associated with morphological alterations of the primary cilium, we proceeded investigating ciliary functions. We focused our analysis on the Shh cascade, because this signaling pathway is generally recognized as the one that is mainly associated with primary cilia (Wheway *et al*, 2018). Shh activation can be detected by measuring the levels of various components of its pathway along the primary cilium or in a ciliary subdomain. In particular, the binding of the ligand Shh to its receptor, protein patched homolog 1 (Ptch1), triggers the accumulation of Smoothened (Smo) within the ciliary membrane with the consequent activation of Gli transcription factors (Fig 3A). We thus analyzed Smo localization in the primary cilium of starved hTERT-RPE-1 cells silenced for MeCP2 and treated with SAG, a Smo agonist able to activate the Shh pathway (Fig 3B). To assess the percentage of Smo-positive cilia, cells were stained for Smo and acetylated α-tubulin; the results demonstrated that MeCP2 depletion significantly impairs the ciliary accumulation of Smo upon Shh stimulation (***$P < 0.001$, two-way ANOVA followed by Bonferroni *post hoc* test). Similar results were obtained in *Mecp2* null MEFs following SAG exposure and a significant difference between cells was evident when the higher dose of SAG was used (*$P < 0.05$, two-way ANOVA followed by Bonferroni *post hoc* test; Fig 3C).

To further support a functional defect of the Shh signaling pathway in *Mecp2*-defective cells, the protein and mRNA levels of the downstream transcription factor Gli1 were measured in cultured MEFs and neurons. Western blots from SAG-treated and untreated cells proved that the lack of Mecp2 impairs Gli1 activation (Fig 3D and E). Indeed, while a significant increase in Gli1 expression was measured both in WT MEFs and in neurons (***$P < 0.001$, two-way ANOVA followed by Bonferroni *post hoc* test), the activation of the Shh pathway did not lead to a significant Gli1 induction in *Mecp2* null cells. Accordingly, Gli1 levels were significantly reduced in SAG-treated *Mecp2* null cells with respect to WT (*$P < 0.05$; **$P < 0.01$, two-way ANOVA followed by Bonferroni *post hoc* test). By analyzing *Gli1* mRNA, we confirmed the defective response to SAG in both *Mecp2* null cells, although cortical neurons exhibited a trend that did not reach statistical significance (Fig 3F and G). Of note, the low expression of *Gli1* mRNA in basal conditions did not permit quantitative analysis.

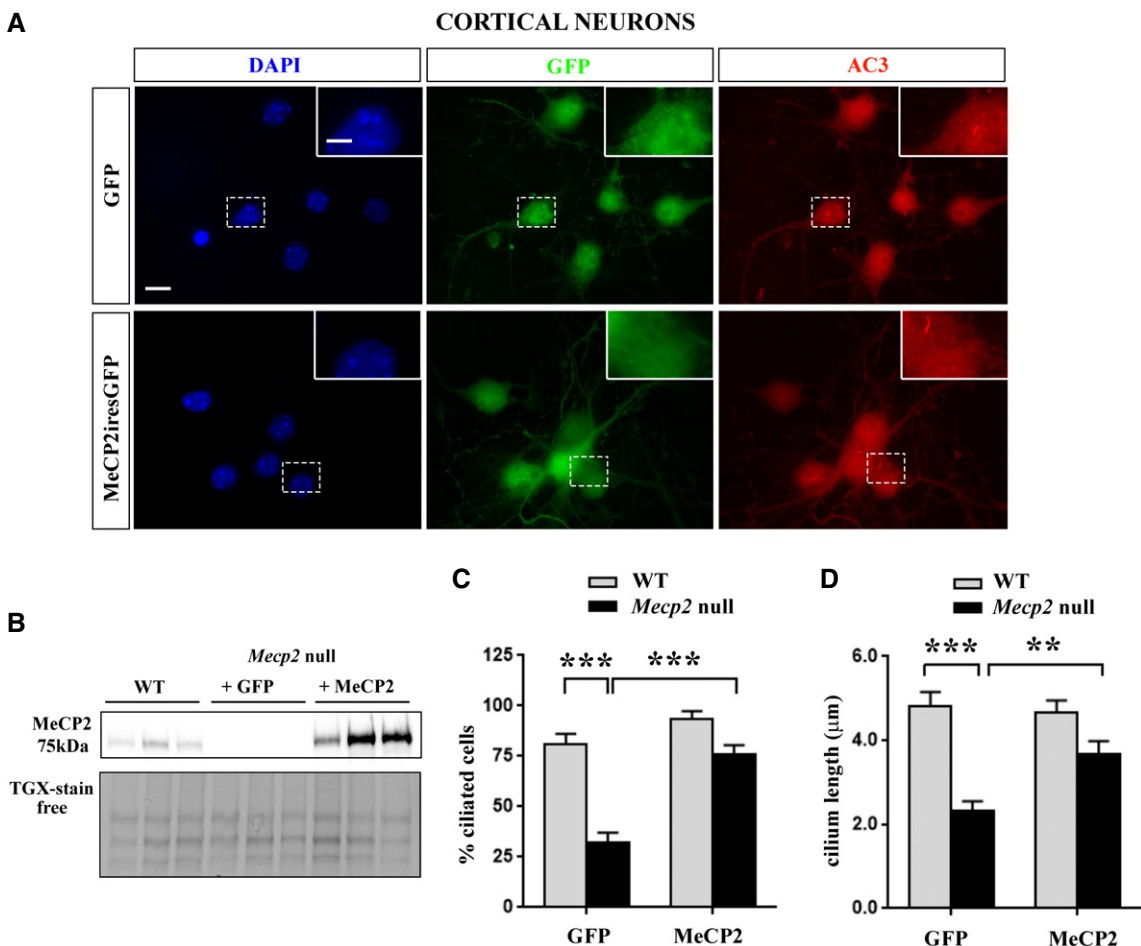

**Figure 2. MeCP2 expression in null neurons rescues primary cilium defects.**

A   Representative immunofluorescence of *Mecp2* null cortical neurons infected at DIV0 with lentiviruses expressing GFP or MeCP2iresGFP. Neurons were fixed at DIV7 and immunostained with anti-AC3 antibody to detect the primary cilium. GFP-positive cells represent infected cells. Scale bar = 20 μm, and 10 μm in the enlarged image.

B   Western blot comparing MeCP2 levels in *Mecp*2 null neurons infected with lentiviruses expressing GFP or MeCP2iresGFP with respect to WT neurons. Data are normalized to total protein content, visualized by a TGX stain-free technology.

C, D   *MECP2* expression in null neurons induced a significant recovery of ciliogenesis. (C) The graph represents the percentage of ciliated cells, counting $n = 50$ WT and $n = 100$ null cells of two different preparations (mean $\pm$ SE, ***$P < 0.001$, two-way ANOVA followed by Bonferroni *post hoc* test). Statistical analysis reported a significant interaction ($F (1, 135) = 8.264$), a significant genotype effect ($F (1, 135) = 37.59$), and a significant infection effect ($F (1, 135) = 26.89$). (D) The graph represents the mean $\pm$ SE of cilium length in WT and *Mecp*2 null neurons infected with GFP- or MeCP2iresGFP-expressing lentiviruses, counting $n = 28$ cells of two different preparations (**$P < 0.01$; ***$P < 0.001$, two-way ANOVA followed by Bonferroni *post hoc* test). Two-way ANOVA reported a significant interaction ($F (1, 103) = 6.405$), a significant genotype effect ($F (1, 103) = 35$), and a significant infection effect ($F (1, 103) = 4.039$).

Source data are available online for this figure.

## *Mecp2* deficiency is associated with a significant shortening of primary cilia in the cerebral cortex and cerebellum

Disruption of neuronal primary cilia can induce defects in neuronal connectivity and reduce intellectual functions (Lee & Gleeson, 2014; Reiter & Leroux, 2017). Thus, we investigated whether *Mecp2* deficiency could also affect ciliogenesis *in vivo*. Considering the involvement of cerebral cortex in RTT, we initially focused on the *Mecp2* null developing cortex (P14). As shown in Fig 4A–C, in knock-out cortices, cilium length was significantly reduced compared to WT controls (*$P < 0.05$, Mann–Whitney test). To identify the most affected cortical layers, the analysis was conducted separately for each layer; besides a significant genotype effect ($F (1, 20) = 45.61$, $P < 0.001$), *post hoc* analysis indicated a significant reduction in cilium length in layers 1, 2/3, and 4 (***$P < 0.001$; **$P < 0.01$; *$P < 0.05$, two-way ANOVA followed by Bonferroni *post hoc* test; Fig 4C).

Although primary cilium is present in most neurons, its role for cerebral cortex development and functioning still remains debated; on the contrary, the primary cilium and the associated Shh pathway are recognized as master players for cerebellum development. In particular, in the cerebellum, Shh activation starts at E17.5, peaks at P5-P8, and then progressively declines (De Smaele *et al*, 2008; Vaillant & Monard, 2009). We thus analyzed cilium length in P7

cerebella, including in the study both *Mecp2* null and heterozygous tissues and discriminating the external (EGL) and the internal (IGL) granular layers (Fig 4D). In good accordance with the more mature state, the WT IGL revealed a longer primary cilium than the WT EGL, while the gender did not play any effect. Importantly, a

significant reduction of cilium length was measured in the two layers of both *Mecp2* null and heterozygous cerebella (**$P < 0.01$; ***$P < 0.001$, Mann–Whitney test), although the magnitude of the effect was higher in knock-out tissues (Fig 4E and F). To support the observed morphological defects, we analyzed Gli1 expression

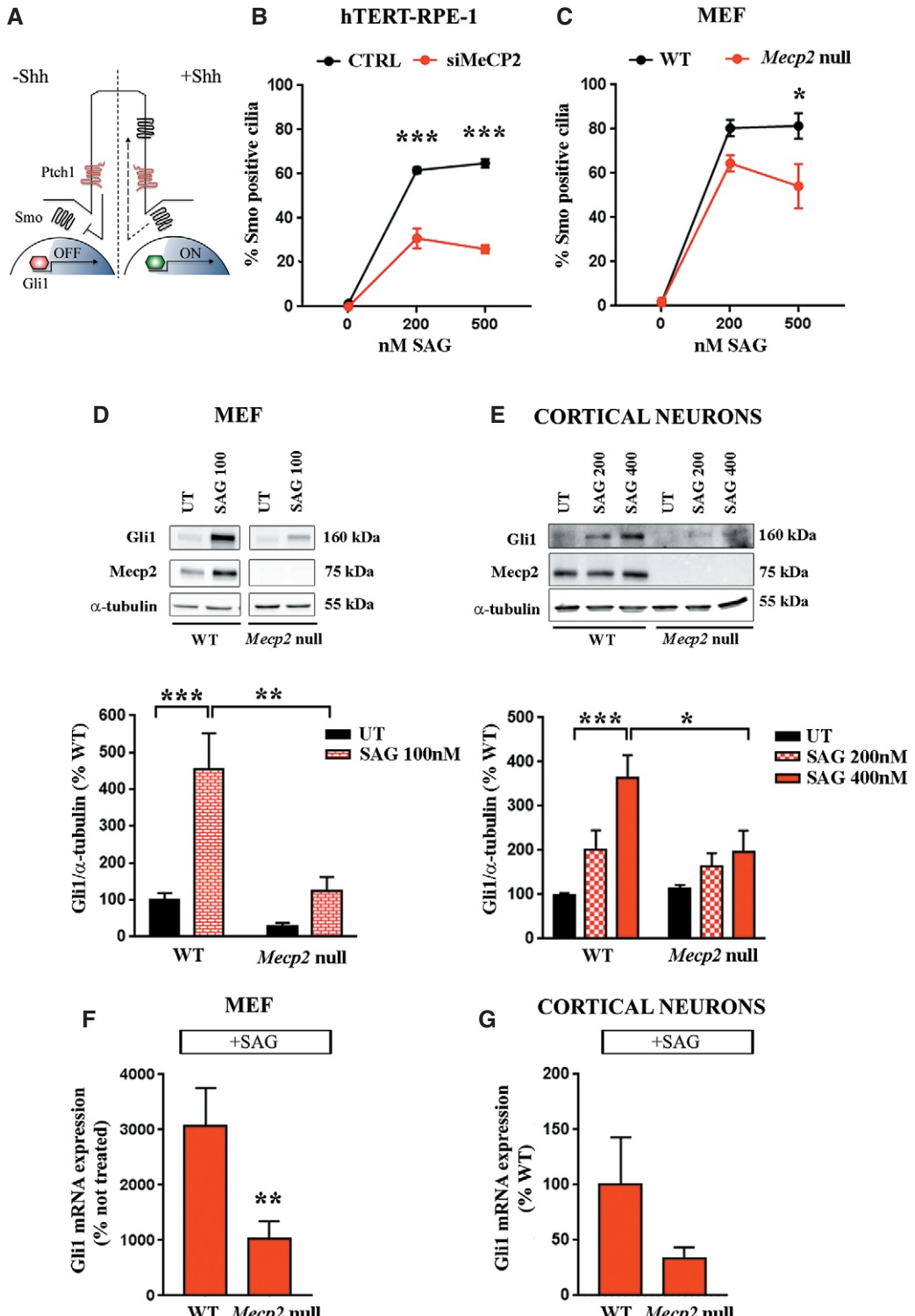

Figure 3.

**Figure 3.** *Mecp2* deletion impairs Shh signaling pathway.

A    Schematic representation of the Shh pathway in the cilium.

B    *MECP2*-silenced or control hTERT-RPE-1 cells were starved for 48 h before treatment with 200 and 500 nM SAG for 24 h. Cells were stained with anti-Smo and anti-acetylated α-tubulin antibodies. The percentages of Smo-positive cilia were calculated. The graph shows the average of three independent experiments (mean ± SE, *** $P < 0.001$, two-way ANOVA followed by Bonferroni *post hoc* test). Two-way ANOVA demonstrated a significant silencing effect ($F_{(1, 12)} = 160$, $P < 0.0001$), a significant SAG treatment ($F_{(2, 12)} = 259.4$, $P < 0.0001$), and a significant interaction ($F_{(2, 12)} = 36.85$, $P < 0.0001$).

C    WT or *Mecp2* null MEF cells were treated with SAG (200 and 500 nM) for 24 h, and the percentages of Smo-positive cilia were calculated. The graph shows the average of three independent experiments (mean ± SE, * $P < 0.05$, two-way ANOVA followed by Bonferroni *post hoc* test). Two-way ANOVA indicated a significant genotype effect ($F_{(1, 12)} = 11.22$, $P < 0.01$) and a significant SAG treatment ($F_{(2, 12)} = 11.7$, $P < 0.0001$).

D    Gli1 protein levels were measured by Western blot in untreated (UT) and 100 nM SAG-treated WT and *Mecp2* null MEF cells ($n = 5/6$). Representative bands of Gli1, Mecp2, and α-tubulin are depicted above the histogram, which shows the mean ± SE of the percentage of Gli1 expression levels, compared to the untreated WT samples (** $P < 0.01$; *** $P < 0.001$, two-way ANOVA followed by Bonferroni *post hoc* test). Two-way ANOVA reported a significant interaction ($F_{(1, 18)} = 6.652$, $P < 0.05$), a significant genotype effect ($F_{(1, 18)} = 15.95$, $P < 0.001$), and a significant treatment ($F_{(1, 18)} = 20.15$, $P < 0.001$).

E    Gli1 protein levels were measured by Western blot in WT and *Mecp2* null neurons, following SAG treatment (200 and 400 nM). Representative bands of Gli1, Mecp2, and α-tubulin are depicted above the histogram, which shows the mean ± SE of the percentage of Gli1 expression levels, compared to the untreated WT samples (* $P < 0.05$, *** $P < 0.001$, two-way ANOVA followed by Bonferroni *post hoc* test). Two-way ANOVA indicated a significant interaction ($F_{(2, 20)} = 3.788$, $P < 0.05$), a significant genotype effect ($F_{(1, 20)} = 4.362$, $P < 0.05$), and a significant treatment ($F_{(2, 20)} = 13.5$, $P < 0.001$).

F, G    *Gli1* mRNA expression levels upon SAG exposure in MEF cells ($n = 7$ WT and $n = 9$ null samples) (F; SAG 100 nM) and cortical neurons ($n = 15$ WT and $n = 12$ null samples) (G; SAG 400 nM). The graphs show the mean ± SE of the fold change derived from at least three independent experiments (mean ± SE, ** $P < 0.01$, two-way ANOVA followed by Bonferroni *post hoc* test).

Source data are available online for this figure.

in the *Mecp2* null cerebellum, which manifests a more severe phenotype. Our results indicated a significant reduction of both Gli1 protein and mRNA (* $P < 0.05$, Student's *t*-test; Fig 4G and H). Gli1 deficiency was supported by a transcriptional reduction of its direct targets, *Cyclin D1* (* $P < 0.05$, Student's *t*-test) and *Cyclin D2* (revealing only a trend of reduction) (Fig 4H; Kenney & Rowitch, 2000).

### Microtubule instability participates in the observed ciliary defects

As already mentioned, ciliary assembly and elongation are negatively influenced by microtubule instability, a defect that is known to occur when MeCP2 is deficient (Delépine *et al*, 2013, 2016). Previously, we have demonstrated that MeCP2 depletion affects the microtubule nucleation potential of the centrosome (Bergo *et al*, 2015); accordingly, RTT patients' fibroblasts are characterized by defective microtubule polymerization and increased HDAC6 levels (Gold *et al*, 2015). By deacetylating α-tubulin, HDAC6 acts as a key driver of primary cilium disassembly and has already been proposed as a possible therapeutic target for ciliopathies (Yu *et al*, 2016).

Considering all above, we initially used tubacin, a potent, selective and cell-permeable HDAC6 inhibitor, to determine whether microtubule stabilization might recover the observed ciliary phenotypes. After starvation, WT and *Mecp2* null MEFs were treated with vehicle (DMSO) or tubacin (1 μM) for 48 h and then stained for acetylated α-tubulin to detect cilia. As depicted in Fig 5A, tubacin significantly restored the percentage of ciliated *Mecp2* null MEFs (* $P < 0.05$, two-way ANOVA followed by Bonferroni *post hoc* test) and *post hoc* analysis confirmed the already-known ciliary defects in *Mecp2* null cells (* $P < 0.05$; ** $P < 0.01$, two-way ANOVA followed by Bonferroni *post hoc* test). Importantly, similar results were obtained when *Mecp2* null cortical neurons were treated with tubacin (Fig 5B) (* $P < 0.05$; ** $P < 0.01$, two-way ANOVA followed by Bonferroni *post hoc* test).

To determine whether the observed morphological rescue could be associated with a functional recovery, we investigated the Shh pathway after tubacin treatment in *MECP2*-silenced hTERT-RPE-1 cells. In particular, we analyzed whether the drug could rescue Smo translocation into the primary cilium upon Shh activation (Fig 5C and D). Control or *MECP2*-silenced hTERT-RPE-1 cells were treated with tubacin or DMSO, and then with SAG (200 nM). The analysis of the percentage of Smo-positive cilia confirmed the defective

**Figure 4.    Primary cilia defects are present in the cortex and cerebellum of *Mecp2* mutant brains.** 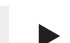

Primary cilium length was measured in the cortex of *Mecp2* null mice and cerebellum of *Mecp2* null and heterozygous mice.

A    Representative immunostaining of Arl13b (red) and γ-tubulin (green) in the P14 mouse cortical layers. Nuclei were stained with DAPI (blue). Scale bar = 10 μm.

B, C    Average cilium length was measured in the cortex (mean ± SE, $n = 4$ WT and $n = 3$ *Mecp2* null) (B; * $P < 0.05$, Mann–Whitney test) and for each cortical layer (C; *** $P < 0.001$; ** $P < 0.01$; * $P < 0.05$, two-way ANOVA followed by Bonferroni *post hoc* test). In (C), cilium length is analyzed in separate cortical layers, while in (B), the values indicate the average of cilium lengths for each layer.

D    Representative immunostaining of Arl13b (green) in the internal (IGL) and external (EGL) granular layers of *Mecp2* null ($n = 3$) and heterozygous (het) cerebella ($n = 3$) at P7, and the corresponding WT ($n = 3$ males and $n = 3$ females). Nuclei were stained with DAPI (blue). Scale bar = 5 μm.

E, F    Histograms show the mean ± SE of cilium length ($n = 3$ male WT, $n = 3$ female WT, $n = 3$ *Mecp2* null and *Mecp2* het; ** $P < 0.01$; *** $P < 0.001$, Mann–Whitney test) in *Mecp2* null (E) and heterozygous (F) mice, compared to the corresponding WT.

G    Gli1 protein levels were measured by Western blot in WT and *Mecp2* null cerebellum ($n = 7$ WT and $n = 6$ *Mecp2* null). Representative bands of Gli1 and α-tubulin are depicted beside the histogram, which shows the mean ± SE of the percentage of Gli1 expression levels, compared to WT animals (* $P < 0.05$, Student's *t*-test).

H    The mRNA expression levels of *Gli1* and its downstream genes (*Cyclin D1* and *D2*) were analyzed in WT and *Mecp2* null cerebella ($n = 8$ WT and $n = 7$ *Mecp2* null). The graph shows the fold change of transcripts compared to WT (mean ± SE, * $P < 0.05$, Student's *t*-test).

Source data are available online for this figure.

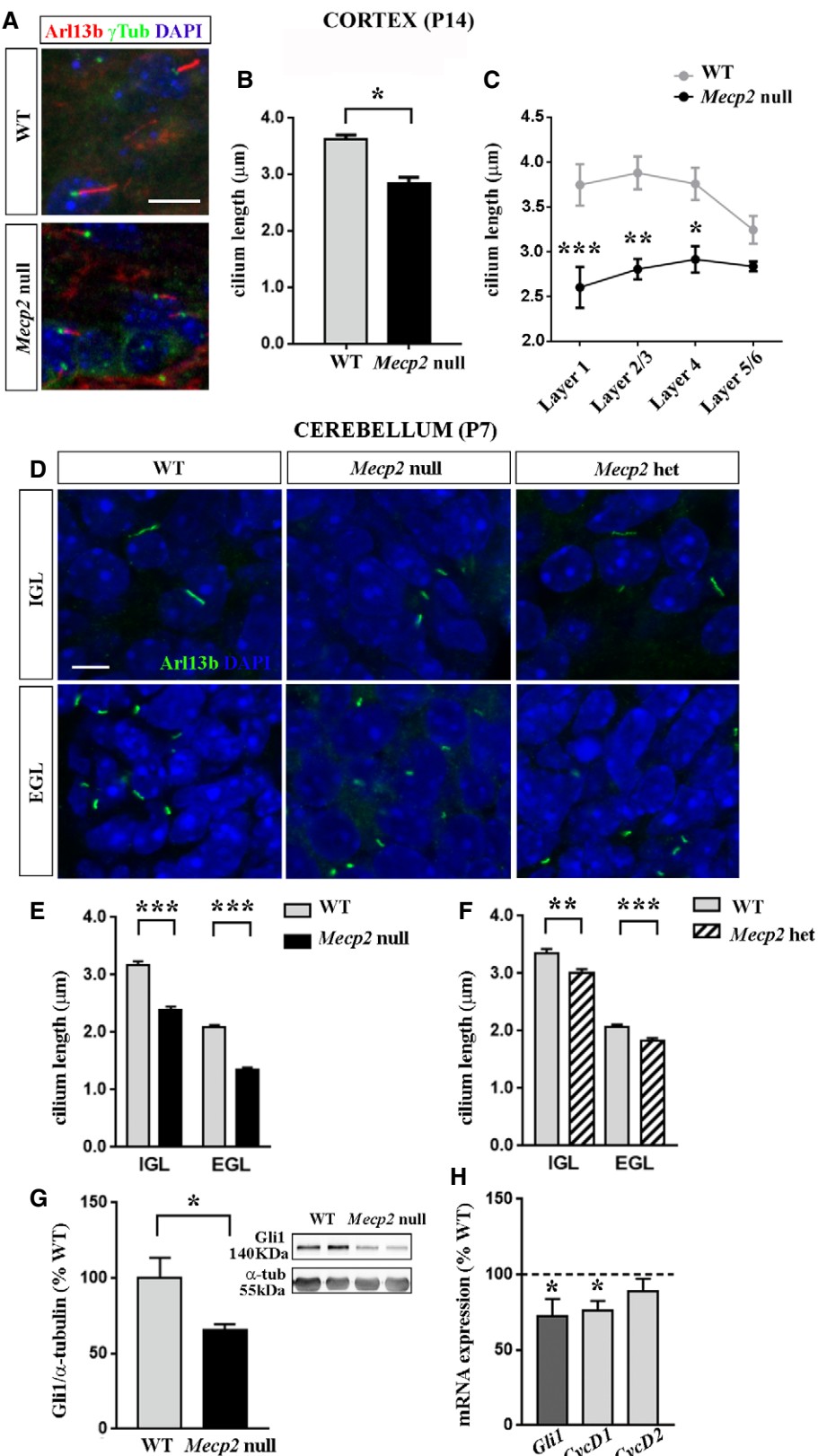

**Figure 4.**

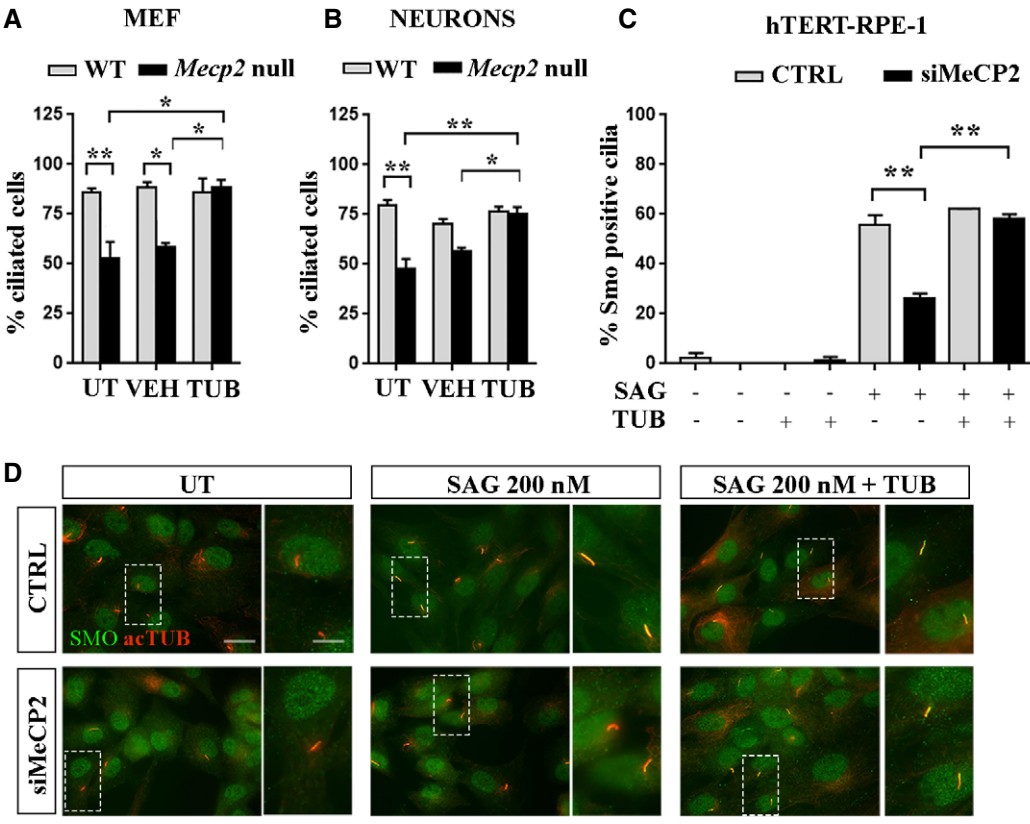

**Figure 5. Tubacin rescues ciliogenesis in MeCP2-deficient cells.**

A   WT and *Mecp2* null MEF cells were starved for 48 h and then treated with the HDAC6 inhibitor tubacin (TUB) (1 µM) for 48 h or with DMSO (VEH) or left untreated (UT). Cells were fixed and stained for anti-acetylated α-tubulin, and cilia were counted. The graph shows the percentage of ciliated cells out of three independent experiments (mean ± SE, *$P < 0.05$; **$P < 0.01$, two-way ANOVA followed by Bonferroni *post hoc* test). Two-way ANOVA reported a significant genotype effect ($F_{(1, 10)} = 17.94$, $P < 0.01$), a significant tubacin treatment ($F_{(2, 10)} = 4.425$, $P < 0.05$), and a significant interaction ($F_{(2, 10)} = 7.266$, $P < 0.05$).

B   Primary cortical neurons were treated at DIV5 with DMSO (VEH) or tubacin (TUB, 1 µM) for 48 h, or left untreated (UT), fixed at DIV7, and stained with anti-AC3 antibody. The graph shows the percentage of ciliated cells obtained from three independent experiments counting $n = 300$ UT cells, $n = 216$ VEH-treated cells, and $n = 267$ TUB-treated cells for each genotype (mean ± SE, *$P < 0.05$; **$P < 0.01$, two-way ANOVA followed by Bonferroni *post hoc* test). Two-way ANOVA reported a significant genotype effect ($F_{(1, 15)} = 23.42$, $P < 0.001$), a significant treatment effect ($F_{(2, 15)} = 11.25$, $P < 0.01$), and a significant interaction ($F_{(2, 15)} = 4.853$, $P < 0.05$).

C   *MECP2*-silenced or control hTERT-RPE-1 cells were starved for 48 h, treated with tubacin (TUB, 1 µM) for 24 h, and then exposed to 200 nM SAG for 24 h in the presence of tubacin. Cells were stained with anti-Smo and anti-acetylated α-tubulin antibodies, and the percentages of Smo-positive cilia were calculated. The graph shows the averages of three independent experiments (mean ± SE, **$P < 0.01$, three-way ANOVA followed by Tukey's *post hoc* test). Statistical analysis reported a significant difference between genotype ($F_{(1, 2)} = 21.47$, $P < 0.001$), a significant SAG effect ($F_{(2, 2)} = 154.5$, $P < 0.0001$), and a significant tubacin effect ($F_{(1, 2)} = 25.18$, $P < 0.001$). In detail, a significant interaction between genotype and SAG treatment ($F_{(2, 2)} = 5.413$, $P < 0.05$) and between SAG and tubacin treatment ($F_{(2, 2)} = 6.939$, $P < 0.01$) was reported, demonstrating respectively a difference between CTRL- and MeCP2-depleted cells in response to SAG and a different SAG effect when cells were treated with tubacin.

D   Representative immunostaining of Smo (green) and acetylated α-tubulin (red) in control (CTRL) and silenced (siMeCP2) hTERT-RPE-1 cells, after stimulation with 200 nM SAG in the presence or absence of the HDAC6 inhibitor tubacin (TUB). Scale bar = 50 µm, and 20 µm in the enlarged image.

Source data are available online for this figure.

activation of the Shh pathway in silenced cells that was rescued by tubacin treatment (**$P < 0.01$, three-way ANOVA followed by Tukey's *post hoc* test).

To reinforce our results, we investigated whether in concomitance with restored ciliogenesis, we could observe an amelioration of neuronal alterations that are commonly detected in *Mecp2* null cells and that are relevant for Rett syndrome. WT and *Mecp2* null neurons were thus treated with tubacin or TC-S 7010, a specific inhibitor of the centrosomal kinase Aurora A. In neurons, Aurora A acts by phosphorylating and activating HDAC6 at the basal body, thus promoting cilium disassembly (Pugacheva *et al*, 2007). Analysis of the cilium length in cortical neurons proved that both the Aurora A inhibitor (7 nM, 48 h) and tubacin (1 µM, 48 h) effectively mediated primary cilium elongation in *Mecp2* null cells (Fig EV2). By Western blot, we reported that TC-S 7010, but not tubacin, maintains the unaffected levels of cytoplasmic acetylated α-tubulin, confirming the more specific action of this inhibitor at the primary cilium level (Pugacheva *et al*, 2007; Fig EV3).

Recent studies suggest that proper formation of neuronal dendrites requires structurally normal primary cilia and a correct

localization of signaling proteins into the cilium, thus linking this organelle to the ability of primary neurons to form correct network connections (Guadiana *et al*, 2013). Therefore, by Sholl analysis we investigated whether tubacin and TC-S 7010 could ameliorate the typical reduction in dendritic complexity in *Mecp2* null neurons (DIV7) (**P < 0.01, Mann–Whitney test; Bedogni *et al*, 2016). Figure 6A, B proves that both molecules significantly increased dendritic branching, thus recovering defective morphology (*P < 0.05; **P < 0.01, one-way ANOVA followed by Dunn's test).

We next analyzed whether HDAC6 inhibition was also effective in ameliorating the synaptic phenotypes of null neurons (Fig 6C–F).

As expected, the analysis of synaptic puncta density in *Mecp2* null DIV14 cortical neurons reported a significant reduction of both pre-synaptic Synapsin1/2 and post-synaptic Shank2 puncta (**P < 0.01; ***P < 0.001, Mann–Whitney test; Fig 6D and E). Interestingly, while TC-S 7010 reverted the defects both at the pre- and the post-synaptic compartments (***P < 0.001, one-way ANOVA followed by Dunn's test), tubacin restored only the density of pre-synaptic puncta (*P < 0.05, by one-way ANOVA followed by Dunn's test). As a measure of functional synapses, we next analyzed the co-localization of Synapsin1/2 with Shank2 puncta. As expected, our data highlighted a defective co-localization in *Mecp2* null neurons compared

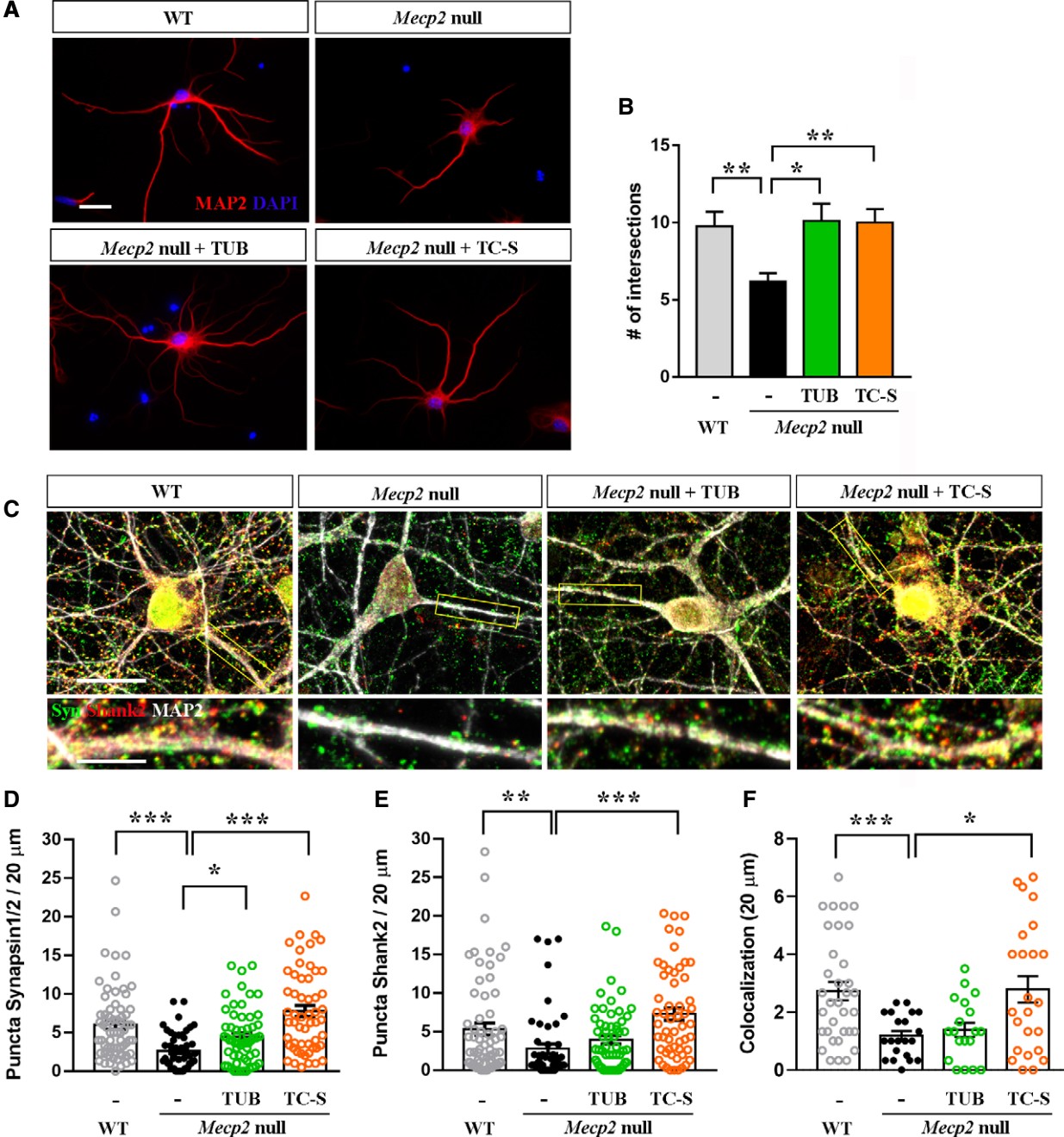

**Figure 6.**

**Figure 6. HDAC6 inhibition restores neuronal defects associated with the lack of Mecp2HDAC6 inhibition restores neuronal defects associated with the lack of Mecp2.**

A    Representative images of MAP2-positive WT and *Mecp2* null neurons (DIV7). At DIV5, neurons were treated with tubacin (TUB, 1 μM for 48 h) and Aurora A inhibitor TC-S 7010 (TC-S, 7 nM for 48 h) or left untreated and morphologically analyzed by Sholl analysis plug-in. Scale bar = 20 μm.

B    The graph reports the mean ± SE of the number of intersections measured 85–145 nm from the soma calculated by Sholl analysis for WT ($n$ = 39 neurons) and *Mecp2* null neurons, when left untreated ($n$ = 57 neurons) or treated with tubacin ($n$ = 53 neurons) or TC-S 7010 ($n$ = 38 neurons). Data from WT and null neurons were compared by Mann–Whitney test (**$P$ < 0.01); drug effects on null cells were analyzed by one-way ANOVA followed by Dunn's *post hoc* test (*$P$ < 0.05; **$P$ < 0.01).

C    Representative images of WT and *Mecp2* null neurons (DIV14) immunostained for MAP2 (white), Synapsin1/2 (green), and Shank2 (red). *Mecp2* null cells were treated at DIV12 with tubacin (TUB, 1 μM for 48 h) or Aurora A inhibitor TC-S 7010 (TC-S, 7 nM for 48 h). Scale bar = 15 μm, and 5 μm in the enlarged image.

D, E   The graphs represent the mean ± SE of the Synapsin1/2 and the Shank2 puncta density in WT ($n$ = 67 neurons) and *Mecp2* null neurons, when untreated ($n$ = 51 neurons) or treated with tubacin ($n$ = 59 neurons) or TC-S 7010 ($n$ = 55 neurons). Neurons derived from at least 6 different biological replicates. Data from WT and null neurons were compared by Mann–Whitney test (**$P$ < 0.01; ***$P$ < 0.001); drug effects on null cells were analyzed by one-way ANOVA followed by Dunn's *post hoc* test (*$P$ < 0.05; ***$P$ < 0.001).

F    The graph depicts the mean ± SE of the number of co-localized pre- and post-synaptic puncta ($n$ = 20 neurons deriving from three different biological samples). Data from WT and null neurons were compared by Mann–Whitney test (***$P$ < 0.001); drug effects on null cells were analyzed by one-way ANOVA followed by Dunn's *post hoc* test (*$P$ < 0.05).

Source data are available online for this figure.

to WT cells (***$P$ < 0.001, Mann–Whitney test), which was rescued only by TC-S 7010 (*$P$ < 0.05, one-way ANOVA followed by Dunn's test; Fig 6F).

## Tubacin restores defective ciliogenesis in RTT patients' fibroblasts

To determine whether the defective ciliogenesis observed in different experimental models would be present also in patients' cells, we took advantage of RTT fibroblasts. To this purpose, we used skin fibroblasts from three RTT female patients characterized by a frameshift mutation (705delG) or a premature stop codon (Q224X and R255X), respectively (Fig 7A). RTT fibroblasts together with 3 independent controls were grown to confluence and then starved to induce growth arrest and cilium formation. Primary cilia were immunolabeled for acetylated α-tubulin to highlight the axoneme and for MeCP2 to distinguish cells expressing the WT allele from those expressing the frameshift mutation or the truncated derivatives. Our data demonstrated that Q244X and R255X fibroblasts expressed preferentially the mutated allele, while less than 20% of cells expressed the frameshift mutation 705delG (Fig 7B and C). This result was confirmed by sequencing the product of an RT–PCR used to amplify the *MECP2* mRNA. Importantly, by counting the percentage of ciliated cells, we revealed that more than 80% of control cells presented a primary cilium; on the contrary, the antenna was present in less than 70% of both fibroblast lines harboring an *MECP2* allele with a nonsense mutation, while the 705delG sample was characterized by a slight but significant reduction in the percentage of ciliated cells, thus confirming a defect in ciliogenesis (*$P$ < 0.05; **$P$ < 0.01; ***$P$ < 0.001, one-way ANOVA followed by Dunnett's *post hoc* test; Fig 7D). Further, in all patients' cells cilium length was affected, indicating primary cilium alteration as a common tract among the analyzed RTT mutations (***$P$ < 0.001, two-way ANOVA followed by Bonferroni *post hoc* test; Fig 7E). Importantly, tubacin treatment (250 nM, 24 h) caused a significant cilium elongation in most of the mutant cells, thus rescuing the structural defect of primary cilium (§$P$ < 0.05, vs. untreated fibroblasts by two-way ANOVA followed by Bonferroni *post hoc* test; Fig 7E).

## Discussion

Rett syndrome is a complex neurodevelopmental disorder characterized by a plethora of clinical alterations affecting both the central and the peripheral system, leading to the manifestation of several symptoms. No treatment is available to combat the primary pathology for any of the *MECP2*-pathies and patients are treated only to alleviate clinical symptoms. The lack of a full comprehension of the molecular mechanisms causing *MECP2*-related pathologies, including Rett syndrome, certainly slows down the identification of effective therapies.

MeCP2 was originally isolated as an epigenetic reader of methylated DNA that represses gene expression mainly through chromatin compaction (Jones *et al*, 1998; Nan *et al*, 2007; Skene *et al*, 2010). However, the proposed gene regulatory role has been expanded beyond gene silencing and chromatin architecture to include transcriptional activation, regulation of mRNA splicing, non-coding RNA maturation and protein synthesis modulation (Bedogni *et al*, 2014; Cheng & Qiu, 2014). While these studies indicate that MeCP2 is a multifunctional protein, they also highlight that we still do not know how many functions are associated with MeCP2 and which ones are relevant to *MECP2*-related disorders. However, it is commonly accepted that the majority of clinical symptoms results from the effect of MeCP2 on gene expression (Bellini *et al*, 2014). The recently identified function of MeCP2 at the centrosome diverges from the above-described roles of the methyl binding protein, since it does not directly impact on gene transcription (Bergo *et al*, 2015). Indeed, most functions of the centrosome rotate around its capacity to nucleate and organize microtubules; in neurons, which represent the cells mostly involved in Rett syndrome, the centrosome plays several functions. It is essential for axon formation (Ge *et al*, 2010) and is generally considered crucial for cell migration and polarization (Kuijpers & Hoogenraad, 2011). Furthermore, in these terminally differentiated cells, the mother centriole of the centrosome migrates toward the cell surface and converts into a basal body, thereby organizing the assembly of the primary cilium (Stearns, 2001).

Considering our previous data suggesting a centrosomal function of MeCP2 (Bergo *et al*, 2015), we found interesting to investigate whether primary cilium might suffer from structural and/or

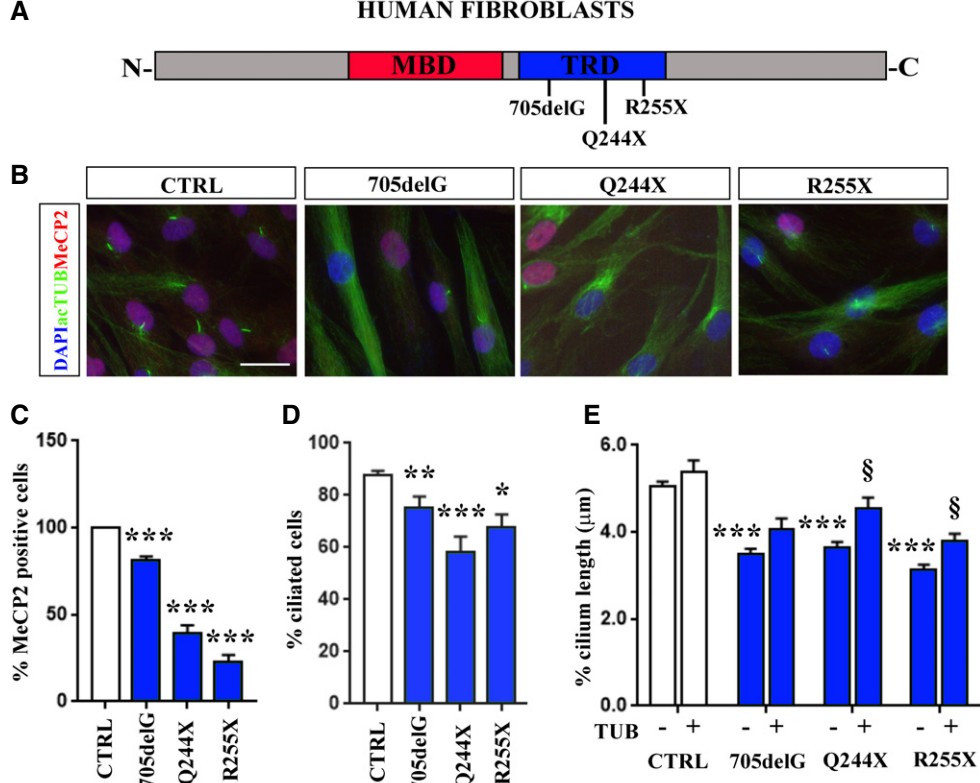

**Figure 7. RTT patients' fibroblasts show defective ciliogenesis, which is rescued by tubacin treatment.**

A    Schematic representation of the MeCP2 protein, with the methyl binding domain (MBD) highlighted in red and the transcriptional repression domain (TRD) in blue. Patients' mutations investigated in this study are indicated.

B    Representative immunostaining of primary cilia and MeCP2 in cultured human fibroblasts from healthy controls (CTRL, n = 3) and RTT patients. Fibroblasts were starved for 24 h and then stained for acetylated α-tubulin (green), MeCP2 (red), and DAPI (blue). Scale bar = 50 μm.

C, D    Histograms indicate (mean ± SE) the percentage of MeCP2-positive cells (C) and the percentage of ciliated cells in control and RTT fibroblasts (D) (*P < 0.05; **P < 0.01; ***P < 0.001, one-way ANOVA followed by Dunnett's post hoc test).

E    The effect of tubacin (TUB) treatment (250 nM, 24 h) on primary cilium length of controls (CTRL) and RTT fibroblasts is reported. The graph indicates the mean ± SE of the cilium length analyzed in untreated (n = 130 CTRL; n = 73 705delG; n = 57 Q244X; n = 68 R255X) and treated cells (n = 25 CTRL; n = 26 705delG; n = 26 Q244X; n = 49 R255X) (***P < 0.001 vs untreated CTRL, §P < 0.05 vs corresponding mutated untreated fibroblasts by two-way ANOVA followed by Bonferroni post hoc test). Statistical analysis reported a significant treatment effect (F (1, 429) = 26.56, P < 0.0001) and a significant genotype (F (3, 429) = 42.85, P < 0.001).

Source data are available online for this figure.

functional alterations in *Mecp2*-deficient cells. Indeed, primary cilia regulate a variety of physiological functions, including metabolism, cell division and differentiation whose relevance is highlighted by a long list of heterogeneous disorders, generally called "ciliopathies" and characterized by defective primary cilia. Notably, minor deviations from the normal range of cilium length characterizing each cell type can lead to pathogenic phenotypes (Avasthi & Marshall, 2012).

In the mature brain, virtually each neuron contains this cellular appendage. Although we still lack a clear comprehension of the main roles of primary cilia in adult neurons, it is generally accepted that their dysfunction is associated with obesity, cognitive impairment and developmental disorders (Sterpka & Chen, 2018). On the contrary, little is known about the functions and consequences of ciliary deficiency in astrocytes (Sterpka & Chen, 2018). Microglia do not display primary cilia (Sipos *et al*, 2018) and we still lack evidence of their presence on mature oligodendrocytes, although they are present on young oligodendrocyte precursor cells (Falcón-Urrutia *et al*, 2015).

Our data demonstrate a structural and functional impairment of primary cilium in any tested MeCP2-defective cell, ranging from silenced human cell lines, *Mecp2* null neurons, astrocytes, and *Mecp2* knock-out or knock-in MEFs. Furthermore, by exogenously expressing MeCP2 in null neurons, we have proved a direct link between the expression of the methyl binding protein and accurate ciliogenesis. Importantly, the presence of primary cilium defects was confirmed in heterozygous RTT patients' fibroblasts characterized by premature stop codons or a frameshift mutation occurring in the TRD and in *Mecp2*$^{Y120D/y}$ MEFs, harboring a missense mutation that mainly affects the capacity of the protein to tightly associate with DNA and chromatin (D'Annessa *et al*, 2018; Gandaglia *et al*, 2018). By inspecting whether the reduction in the primary cilium length is proportionally related to a reduction in soma size, we found that *Mecp2* null neurons exhibit the well-known reduction in the soma area that is not present in RTT patients' fibroblasts; however, in both cell types

no correlation exists between primary cilium length and soma size (Appendix Fig S1).

We hypothesize that the transcriptional functions of MeCP2 contribute to the observed ciliary defects, although an involvement of its centrosomal functions cannot be excluded. Indeed, we have recently shown that in the absence of MeCP2, microtubule polymerization from centrosomes is impaired (Bergo *et al*, 2015). In good accordance, a reduction in microtubule stability associated with decreased α-tubulin acetylation, caused by increased levels of HDAC6, was reported (Xu *et al*, 2014; Gold *et al*, 2015; Delépine *et al*, 2016; Landucci *et al*, 2018). Real-time RT–PCR suggested that a post-transcriptional mechanism is involved in the observed HDAC6 upregulation (Delépine *et al*, 2016) and our unpublished data confirm that *Hdac6* mRNA levels are not perturbed in the *Mecp2* null brain. Of note, although the molecular mechanisms causing this aberrant HDAC6 activity remain unknown, it has already been reported that miRNA dysregulation can lead to HDAC6 overexpression (Mansini *et al*, 2018) and *Mecp2* deficiency affects the transcription of several miRNAs (Ip *et al*, 2018 and references therein). These results suggest a possible mechanism involved in the observed ciliary defect. In fact, it is generally recognized that by directly affecting microtubule stability, HDAC6 can control ciliary length (De Diego *et al*, 2014). By demonstrating that HDAC6 inhibition rescues ciliogenesis in *MECP2*-mutated cells with a concomitant amelioration of their typical neuronal dysfunctions, we proved that microtubule instability participates in the observed phenotypes, thereby offering a possible therapeutic target. Further, we demonstrated that TC-S 7010, that in postmitotic cells preferentially affects HDAC6 activity in the primary cilium through the inhibition of Aurora A (Pugacheva *et al*, 2007; Korobeynikov *et al*, 2018), is significantly more effective than tubacin, which inhibits cytoplasmic HDAC6. Altogether these results support the validity of a pharmacological treatment acting selectively on the stability of primary cilium and indicate that the stabilization of primary cilium can lead to an improvement in neuronal complexity and synaptic alterations, in accordance with the role of this organelle in neuronal development and synaptic plasticity (Rhee *et al*, 2016). Although the exact mechanisms associating primary cilium with dendrites are not fully understood, recent evidence suggests that neuronal dendritogenesis depends on the formation of structurally and functionally normal cilia (Guadiana *et al*, 2013). However, although we suggest that the observed amelioration of the number of synapses and dendrites is associated with rescued ciliogenesis, we cannot exclude the intervention of another mechanism independent of cilia formation.

We reinforced our *in vitro* observations demonstrating defective ciliogenesis in *Mecp2* null and heterozygous mouse brains. Indeed, we have reported ciliary defects in *Mecp2* null cortices (P14) and in *Mecp2* null and heterozygous developing cerebella (P7). The occurrence of the defect also in heterozygous brains is particularly relevant to Rett syndrome. Further, most of the cells has reduced cilium length without showing a bimodal distribution, although almost 40% of them express Mecp2, thereby suggesting the involvement of a non-cell-autonomous mechanism.

Since the Shh signaling pathway still represents the best-established primary cilium-associated pathway, we analyzed whether MeCP2 dysfunctions might affect it. Our *in vitro* results proved a functional impairment of the primary cilium in human and murine MeCP2-defective samples. Further, our *in vivo* data indicate that *Mecp2* null cerebella are characterized by reduced levels of Gli1 and a defective expression of other members of the same pathway, opening up new perspectives for the comprehension of the pathogenic mechanisms contributing to RTT and their treatment.

Conditional disruption of ciliogenesis in the mouse cortex and hippocampus impacts on learning and memory and modifies paired-pulse response of hippocampal neurons (Berbari *et al*, 2013), and many genes associated with psychiatric disorders affect ciliogenesis *in vitro* (Marley & von Zastrow, 2012). Furthermore, multiple ciliopathies, such as Joubert syndrome and Bardet–Biedl syndrome, present both neurodevelopmental and cognitive deficits. As a matter of facts, loss-of-function mutations in ciliary genes often results in cerebellar hypoplasia, and mice with conditional mutations disrupting the Shh pathway exhibit defects in cerebellar foliation and granule cell proliferation (Corrales *et al*, 2006). Interestingly, previous studies of magnetic resonance imaging (MRI) have reported a reduction in cerebellar volume both in RTT patients and in *Mecp2* null mouse models (Allemang-Grand *et al*, 2017; Shiohama *et al*, 2019). These results, together with the well-recognized association of cerebellar dysfunctions with autistic-like behaviors and ataxia (Tsai *et al*, 2018), prompt a detailed description of the development of *Mecp2* null cerebellum and whether *Mecp2* null brains manifest alterations in cortico-cerebellar motor circuitries. In the future, we will also investigate in mouse models of RTT the therapeutic potential of rescuing ciliogenesis or the associated signaling pathways.

To conclude, our data demonstrate for the first time that MeCP2 influences proper formation and functioning of primary cilium, which in turn might contribute to the occurrence of central and metabolic symptoms typical of Rett syndrome. Thus, although we do not want to define Rett syndrome as a ciliopathy, we consider important to highlight that it shares several symptoms with these disorders, such as neurodevelopmental alteration, intellectual disability, autistic features, ataxia, seizures, respiratory abnormalities, obesity, hypotonia and skeletal bone defects (Hagberg *et al*, 2002; Novarino *et al*, 2011). We therefore suggest the importance of future studies testing whether novel pharmacological approaches effective for ciliopathies could be re-directed on Rett syndrome.

## Materials and Methods

### Drugs

Shh small molecule agonist (SAG) was purchased from Sigma-Aldrich (SML1314) and dissolved in sterile water. MEFs and hTERT-RPE-1 cells were treated with SAG (100 nM for MEFs and 200 nM and 500 nM for hTERT-RPE-1 cells) for 24 h after serum starvation with low serum medium (0.5% fetal bovine serum). Neurons were treated with 200 nM and 400 nM SAG for 48 h. Tubacin (SML0065, Sigma-Aldrich) was dissolved in DMSO. MEFs, neurons, and hTERT-RPE-1 cells were treated with 1 μM tubacin for 48 h, whereas fibroblasts were treated with 250 nM tubacin for 24 h in low serum medium. Aurora A inhibitor I (TC-S 7010, Selleckchem) was dissolved in DMSO, and a dose of 7 nM was used on neurons for 48 h.

## Plasmids

To silence *MECP2*, hTERT-RPE-1 cells were transfected with the already-described siMeCP2#DH4 (sense, 5′-GGAAAGGACUGAA GACCUGUU-3′) or, as corresponding control, the scrambled siRNA (sense, 5′-UAGCGACUAAACACAUCAA-3′), both purchased from Dharmacon (Bergo *et al*, 2015). Neurons (DIV0) were infected with the already-described lentiviruses expressing MeCP2iresGFP or exclusively GFP as control (Stefanelli *et al*, 2016).

## Animals

As lengthily described (Cobolli Gigli *et al*, 2016), the *Mecp2* null mouse strain, originally purchased from the Jackson Laboratories, was backcrossed and maintained on a clean CD1 background. These mice recapitulate the typical phenotype of C57BL/6 mice, with the advantage of having a larger progeny and minor risk of litter cannibalization (Cobolli Gigli *et al*, 2016). *Mecp2* null mouse genotype was determined by PCR protocol on genomic DNA purified from tails using the following primers: 5′-ACCTAGCCTGCCTGTACTTT-3′ as forward primer for null allele; 5′-GACTGAAGTTACAGATGG TTGTG-3′ as forward primer for wild-type allele; and 5′-CCACCCTC CAGTTTGGTTTA-3′ as common reverse primer. *Mecp2*$^{Y120D/y}$ mouse genotype was assessed using the following primers: 5′-CAG GGCCTCAGAGACAAGC-3′ as common forward primer; 5′-GCAGA TCGGCCAGACTTCC-3′ as common reverse primer; and 5′-GGGT TAATTGATATCCAATTGGGATCC-3′ as reverse primer for knock-in allele. For mouse embryonic fibroblasts (MEFs) and neuronal/astrocytic cultures, wild-type (WT) and *Mecp2* null embryos or pups were generated by mating *Mecp2* heterozygous females with WT male mice. The day of vaginal plug was considered E0.5 and primary neurons were prepared from E15.5 embryos, while astrocytic cultures were prepared from post-natal day 2 (P2) pups. E13.5 embryos were used for MEF preparation. WT and *Mecp2* null mice (P7) derived from three separate litters were used to collect cerebella. Mice were rapidly decapitated, brain was removed, cerebellum was isolated and immediately frozen on dry ice and conserved at −80°C until analysis. P7 and P14 WT, *Mecp2* null and heterozygous mice derived from three litters were used for immunostaining experiments and confocal analysis.

Animals were housed in a temperature- and humidity-controlled environment in a 12-h light/12-h dark cycle with food and water *ad libitum*. All procedures were performed in accordance with the European Union Communities Council Directive (2010/63/EU) and Italian laws (D.L.26/2014). Protocols were approved by the Italian Minister for Scientific Research and by the San Raffaele Scientific Institutional Animal Care and Use Committee in accordance with the Italian law.

## Cell cultures

### Mouse embryonic fibroblasts (MEFs)

Mouse quiescent embryonic fibroblasts were prepared from wild-type and *Mecp2* null E13.5 embryos, as previously described (Bergo *et al*, 2015). Harvested embryos were placed in a 10-cm dish with PBS. Heads were collected for genotyping and the inner organs were discarded. Each embryo was minced with forceps and incubated at 37°C for 30 min in 0.25% trypsin/EDTA (Life

Technologies) in PBS. After trypsin removal, the tissue was re-suspended in Dulbecco's modified Eagle's medium (DMEM; Sigma-Aldrich) with 10% fetal bovine serum (FBS; Gibco) and antibiotics by gentle trituration with a Pasteur pipette, and the supernatant was transferred to T25 flasks. MEFs were maintained in DMEM supplemented with 10% FBS, L-glutamine (Sigma-Aldrich), and 1% penicillin/streptomycin (P/S; Sigma-Aldrich) and grown at 37°C with 5% $CO_2$. MEFs were starved for 24 or 48 h with low serum medium (0.5% FBS) and eventually treated with drugs (SAG and/or tubacin).

### Immortalized human retinal pigment epithelial (hTERT-RPE-1)cells

hTERT-RPE-1 (kindly provided by Dr. Storchova, Max-Planck Institute) cells were maintained in DMEM/F12 (Dulbecco's modified Eagle's medium: 1:1 mixture of DMEM and Ham's F-12 Nutrient Mixture; 15 mM HEPES; Sigma-Aldrich); medium was supplemented with 10% FBS, L-glutamine and 1% P/S. Cells were grown at 37°C with 5% $CO_2$. For siRNA transfection, 20 nM siRNA oligonucleotide targeting *MECP2* or control siRNA was transfected into RPE-1 cells using Lipofectamine™ RNAiMAX (Invitrogen). Silenced cells were processed 96 h post-transfection. After starvation, cells were treated with 1 μM tubacin for 24 h and then exposed to 200 nM SAG.

### Cortical neuronal cultures

Primary cortical neurons were prepared from WT and *Mecp2* null mouse embryos at 15.5 days. Embryos were sacrificed by decapitation and brains were removed under a microscope and immersed in ice-cold Hank's Buffered Salt Solution (HBSS; Life Technologies). Meninges were removed, and cerebral cortex was rapidly dissected and maintained in cold HBSS. Tissues were incubated with 0.25% trypsin/EDTA for 7 min at 37°C and the digestion was blocked with 10% FBS in DMEM (Life Technologies). Cortices were then mechanically dissociated by pipetting in Neurobasal medium, containing 10% FBS, 2% B27 (Life Technologies), 1% L-glutamine (Sigma-Aldrich) and 0.5% P/S (Sigma-Aldrich). For immunofluorescence experiments, neurons were seeded on coated glass coverslip (1 mg/ml poly-L-lysine hydrobromide; Sigma-Aldrich) in 24-well dishes at the density of 50,000 cells/well or at low density (20,000 cells/well) for morphological analysis.

### Cortical astrocyte cultures

Primary astrocyte cultures were prepared from cerebral tissue of P2 wild-type and *Mecp2* null mice. Mice were decapitated and brain was removed. Meninges were gently stripped off from the individual cortical lobe and cortices were isolated and immersed in HBSS containing 10 mM HEPES. Tissues were then immersed in 0.25% trypsin/EDTA for 30 min at 37°C and then mechanically dissociated in Dulbecco's modified Eagle's medium (DMEM; Life Technologies) containing F10 Nutrient (Life Technologies), 10% FBS and 0.5% P/S. The resulting cells were centrifuged at 400 *g* for 7 min, re-suspended in culture medium, and plated in poly-L-lysine (15 μg/ml)-coated flasks. At DIV4, flasks were shaked at 200 rpm for 8 h at 37°C to eliminate residual microglia, and the medium was replaced with fresh culture medium. Cells were incubated in a humidified incubator at 37°C and 5% $CO_2$ until DIV11 when confluence was reached. Astrocytes were detached by 0.25% trypsin/EDTA in HBSS and seeded onto glass coverslips

at a density of 20,000 cells/cm$^2$ for immunofluorescence analysis. A starvation protocol was applied, by incubating cells with medium without FBS for 24 h. Glial cultures contained 95% GFAP$^+$ astrocytes and 5% Iba1$^+$ microglia and no neuron, as indicated by the lack of MAP2 reactivity.

### Human fibroblasts

Human control fibroblasts from healthy individuals (GM00037, GM00969, and GM01651) and fibroblasts carrying 705delG (GM07982) and Q244X (GM16548) mutations were provided by Coriell Institute of Medical Research. Human fibroblasts carrying R255X mutation (R_FFF2238) were obtained from Telethon Network of Genetic Biobank. Cells were maintained in DMEM, containing 10% FBS, 1% L-glutamine, and 1% P/S, and they were grown at 37°C with 5% CO$_2$. Fibroblasts were starved for 24 h with low serum medium (0.5% FBS) and then used for immunofluorescence and Western blot.

### Immunoblot analysis

hTERT-RPE-1, MEFs, and primary cortical neurons were lysed in ice-cold lysis buffer (50 mM Tris–HCl, pH 7.4, 150 mM NaCl, 5 mM EDTA, 1% NP-40, and 1X complete EDTA-free protease inhibitor cocktail [Roche Diagnostic]) for 30 min on ice and then centrifuged for 30 min at 10,000 $g$ at 4°C. Cerebella were lysed in ice-cold lysis buffer for 30 min, sonicated on ice for 10 s at 30 amplitude, and then clarified by centrifugation for 30 min at 10,000 $g$ (4°C). Protein concentrations were measured by the Bradford assay (Bio-Rad). Equal amounts of protein lysates were separated on polyacrylamide gel for SDS–PAGE and proteins were blotted onto a nitrocellulose membrane using a semidry transfer apparatus (Trans-blot SD; Bio-Rad). Membranes were incubated 1 h in blocking solution (Tris-buffered saline containing 5% nonfat milk and 0.1% Tween-20, pH 7.4) and then incubated overnight (4°C) with the primary antibody diluted in blocking solution. The following primary antibodies were used: anti-MeCP2 (M9317; Sigma-Aldrich, 1:1,000), anti-Gli1 (L42B10; Cell Signaling, 1:1,000), anti-acetylated α-tubulin (32-2700; Invitrogen, 1:2,000), anti-α-tubulin (T5168; Sigma-Aldrich, 1:10,000), and anti-GAPDH (SAB1405848; Sigma-Aldrich, 1:10,000). After three washes in TBS-T (Tris-buffered saline containing 0.1% Tween-20, pH 7.4), blots were incubated with HRP-conjugated secondary antibody (1:10,000 in 5% milk in TBS-T) for 1 h at room temperature. The immunocomplexes were visualized by using the ECL substrate (GeneSpin) and Uvitec system. Band density measurements were performed using Uvitec software. Results were normalized to α-tubulin (MEFs, neurons, and cerebella), to GAPDH (hTERT-RPE-1) or to total protein content (neurons) visualized by a TGX stain-free method (Bio-Rad).

### RNA extraction and quantitative real-time PCR

Total RNA from MEFs, primary neurons and cerebella was extracted using PureZOL (Bio-Rad) and quantified using a NanoDrop spectrophotometer. After having assessed RNA integrity by agarose electrophoresis, RNA was reversely transcribed using the RT2 First Strand Kit (Qiagen) as instructed by the manufacturer. The resulting cDNA was used as a template for qRT–PCR with SYBR Green Master Mix (Applied Biosystems) with designated primers for *Gli1*: Forward

Primer: GGAAGGGGACATGTCTAGC; Reverse Primer: AATGGCA CACGAATTCCTTC; *CyclinD1*: Forward Primer: CCCCAA CAACTTCCTCTCCT; Reverse Primer: CCAGCCTCTTCCTCCACTT; and *CyclinD2*: Forward Primer: GAGTGGGAACTGGTAGTGTTG; Reverse Primer: CGCACAGAGCGATGAAGGT. Melting curve showed a single product peak, indicating good product specificity. *HPRT* and *RPL-13* served as internal standards, and fold change in gene expression was calculated using the 2(−delta $C_t$) method.

### Immunofluorescence

#### Immunofluorescence on cellular cultures

Cells seeded on glass coverslips were fixed with 4% paraformaldehyde in PBS for 30 min at room temperature and then washed three times with 10 mM PBS. For the analysis of synaptic puncta, neurons (DIV14) were fixed for 8 min with 4% paraformaldehyde dissolved in PBS with 10% sucrose. Cells were permeabilized at 4°C in PBS containing 0.1% Triton X-100 for 3 min and blocked with 10% FBS in PBS for 15 min. Then, cells were incubated with primary antibody overnight (4°C). Immunofluorescence staining was done with anti-MeCP2 (M9317, Sigma-Aldrich, 1:100), anti-γ-tubulin (T5326, Sigma-Aldrich, 1:500), anti-acetyl-α-tubulin (lys40, 5335, Cell Signaling, 1:500), anti-Smoothened (ab38686, Abcam, 1:200), anti-adenylate cyclase type 3 (AC3, ab125093, Abcam, 1:1,000), anti-MAP2 (8707, Cell Signaling, 1:1,000), anti-Synapsin1/2 (ab106006, Synaptic System, 1:500), and anti-Shank2 (ab162211, Synaptic System, 1:300). Cells were then incubated with Alexa Fluor 488-, Alexa Flour 546-, or Alexa Fluor 647-conjugated secondary antibodies (Life Technologies) for 1h and then washed with PBS. DNA was stained with DAPI solution (1:1,000 in PBS) for 10 min, and slides were mounted with Fluoromount-G reagent (eBioscience). For the quantitative evaluation of the percentage of MeCP2-expressing cells, ciliated cell, cilium length, and ciliary Smo localization, images were acquired from at least three coverslips for each experimental group with an epi-fluorescence microscope by Nikon and analysis was conducted using ImageJ software. To analyze dendritic complexity, isolated MAP2-positive neurons at DIV7 were acquired under an epi-fluorescence microscope by Nikon with a 40× objective and neuronal arborization was analyzed by using a Sholl analysis plug-in distributed with ImageJ software. For the analysis of synaptic puncta, 144.88 × 144.88 µm$^2$ Z-stack images (1,024 × 1,024 pixel resolution, 8-bit grayscale depth) were acquired at 1× digital zoom using a 63× oil-immersion objective by laser-scanning confocal microscope (Zeiss LSM 800, Carl Zeiss) with a step size of 0.3 µm. For each dataset acquisition, parameters (offset background, digital gain, and laser intensity) were maintained constant among different experiments. By ImageJ software, maximum intensity projection images were converted to binary images and processed with a fixed threshold for each channel acquired. The puncta density was calculated by counting only puncta lying along manually selected ROIs within 20 µm of 3 primary branches/neuron. Only puncta with a minimum size of 0.16 µm$^2$ were counted using *Analyze Particles*. To assess puncta co-localization of pre- and post-synaptic markers, the ImageJ Plugin *Colocalization highlighter* was run on each Z-stack image acquired. Co-localized puncta were quantified in manually selected ROIs of the binary mask created from the maximum intensity projection.

### Immunofluorescence on brain sections

Mice were anesthetized by using a mixture of Rompun/Zoletil and transcardially perfused with 4% paraformaldehyde in PBS. Brains were post-fixed overnight at 4°C and slowly dehydrated by using increasing concentrations of sucrose in PBS at 4°C. Brains were embedded with Tissue-TEK (O.C.T, Sakura Finetek), frozen in isopentane, and stored at −20°C. Sectioning was then performed with a cryostat. Antigen retrieval was applied before immunostaining, and then, sections were permeabilized using 0.3% Triton X-100 in PBS for 30 min and quenched with 0.1 M glycine for 30 min. Sections were incubated with mouse anti-Arl13b (ab136648, Abcam, 1:300) and/or rabbit anti-ɣ-tubulin (T5326, Sigma-Aldrich, 1:1,000) primary antibodies, overnight at 4°C, and then with secondary antibodies for 1 h at RT in a solution of 0.2% gelatin, 300 mM NaCl, and 0.3% Triton X-100 in PBS. DNA was stained in the last wash using DAPI for 30 min at RT and mounted using ProLong antifade reagent (Thermo Scientific). Imaging was performed using a Leica TCS SP5-AOBS 5-channel confocal system (Leica Microsystems) equipped with a 405-nm diode, an argon ion, and a 561-nm DPSS laser. Fixed cells were imaged using a PLAPO 40X/1.2 NA oil-immersion objective. All the images were analyzed by using Bitplane Imaris software.

### Statistical analysis

Data are expressed as mean ± SEM and were analyzed using GraphPad Prism software 7.0. The percentage of ciliated cells and cilium length in cultured cells and morphological data in neurons were obtained from cells derived from three separate experiments. For the pharmacological experiments, cells derived from the same batch were subdivided among all the experimental groups. For *ex vivo* analyses, we used animals derived from at least three different litters, in order to exclude maternal effect on the observed phenotypes. Image analyses were performed by a researcher blind to the genotype and/or treatment. In order to use the correct statistical test, we first evaluated the normal distribution of data by applying the D'Agostino–Pearson normality test. Student's *t*-test or Mann–Whitney test was used for the statistical analysis when WT samples were compared to *Mecp2* mutant samples. One-way ANOVA was used to statistically compare the effects of tubacin and TC-S 7010 on *Mecp2* null neurons. Data from the experiment on neuronal cultures at different DIVs, from the experiment on infected neurons, and from Smo-localization experiments and pharmacological experiments were analyzed by two-way ANOVA. Three-way ANOVA was used to statistically analyze Smo-localization data in h-TERT-RPE-1 cells following pharmacological treatment. When there was a significant effect of treatment or genotype, or a significant interaction between the variables (two-way or three-way ANOVA), appropriate *post hoc* test was applied. A *P*-value < 0.05 was considered significant.

**Expanded View** for this article is available online.

### Acknowledgements

This work was mainly supported by the Italian parents' association "ProRETT Ricerca" to N.L. We are grateful to all members of N.L. and C.K.N laboratories for helpful discussions. The RTT fibroblasts were obtained by cell lines and DNA bank of Rett syndrome, X-linked mental retardation and other genetic diseases, member of the Telethon Network of Genetic Biobanks (project number no. GTB12001), funded by Telethon Italy, EuroBioBank network and the "Associazione italiana Rett O.N.L.U.S."

### The paper explained

#### Problem

Mutations in the *MECP2* gene are responsible for a large spectrum of neurological disorders mostly affecting females. Among these, Rett syndrome represents the best defined and frequent condition. No cure is currently available for *MECP2* disorders, and ongoing treatments are usually based on supportive therapies. The attainment of efficient therapies requires a better understanding of the functions exerted by MeCP2 beyond its well-known role as a transcriptional regulator.

#### Results

We demonstrate that MeCP2 is involved in the correct formation and functioning of primary cilium, a cellular organelle that emerges from the surface of every mammalian cells and is altered in a set of diseases defined "ciliopathies" that share some clinical traits with Rett syndrome. These defects have been observed in cultured cells defective for MeCP2, in the brain of transgenic mice modeling the disease and in Rett patients' fibroblasts. We have rationally designed pharmacological interventions that are able to rescue the structure and function of primary cilia in MeCP2-defective cells. Importantly, these drugs have the capacity to recover neuronal defects typical of Rett syndrome.

#### Impact

By demonstrating the involvement of MeCP2 in ciliogenesis, we highlight a novel therapeutic target for *MECP2* disorders. Although we do not want to define Rett syndrome as a ciliopathy, we highlight the importance to considering whether novel pharmacological approaches effective for ciliopathies could be re-directed for Rett syndrome.

### Author contribution

AF, ES, MP, EA, MMV, BL, AB, FB, and VDC conducted the experiments. AF and NL designed the study, prepared the figures, and wrote the manuscript. ES, MP, FDC, and CK-N revised the manuscript and assisted in interpreting and discussing some results.

### Conflict of interest

The authors declare that they have no conflict of interest.

### For more information

(i)   http://biobanknetwork.telethon.it/
(ii)  http://www.eurobiobank.org/
(iii) https://www.coriell.org/1/Browse/Biobanks
(iv)  https://prorett.org/; https://www.rettsyndrome.org/; https://www.biorender.com

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
