## [Review Process File · EMBO Molecular Medicine]

MECP2 mutations affect ciliogenesis: a novel perspective for Rett syndrome and related disorders

Angelisa Frasca, Eleonora Spiombi, Michela Palmieri, Elena Albizzati, Maria Maddalena Valente, Anna Bergo, Barbara Leva, Charlotte Kilstrup-Nielsen, Federico Bianchi, Valerio Di Carlo, Ferdinando Di Cunto and Nicoletta Landsberger

Review timeline:

Submission date:	3rd Jan 2019
Editorial Decision:	10th Jan 2019
Authors' correspondence:	24th Jul 2019
Editor's correspondence:	25th Jul 2019
Authors' correspondence:	30th Jul 2019
Editor's correspondence:	6th Aug 2019
Revision received:	9th Aug 2019
Editorial Decision:	1st Oct 2019
Revision received:	30th Jan 2020
Editorial Decision:	9th Mar 2020
Revision received:	19th Mar 2020
Accepted:	26th Mar 2020

Editor: Céline Carret

Transaction Report:

1st Editorial Decision

10th Jan 2019

Thank you for submitting your manuscript "MECP2 mutations affect ciliogenesis: a novel perspective for Rett syndrome and related disorders" to EMBO Molecular Medicine. I have now read your manuscript and discussed it with the other members of our editorial team. I regret that we find that the manuscript not well suited for publication in EMBO Molecular Medicine and have therefore decided not to proceed with peer review.

We recognise the value of associating MecP2 loss-of-function (LOF) mutations with ciliopathies as a novel take on Rett syndrome and putative therapies. Following up from your initial discovery that a small fraction of MecP2 associates with centrosome, you here report that MecP2 LOF mutations result in aberrant ciliogenesis in mice and human cells (including neurons), in patients' cells and in vivo in the brain of MecP2 null mice. Mechanistically, Shh signalling is affected in vitro and in vivo and microtubule instability participates in the pathology. Tubacin treatment on primary cortical neurons from MecP2 null mice restore microtubule stability and Shh signalling, leading to ciliogenesis rescue. While the study is interesting and novel, unfortunately you stop short at showing that HDAC6 inhibition could be a therapeutic strategy in vivo in MecP2 null mice or in the novel MecP2(Y120D) KI mice that you published recently, or at least in the patients' cells. As EMBO Molecular Medicine focuses on translational advances, such a proof-of-concept is needed for us to send the paper out for review. Should you add these data though, I'll be really happy to send the paper out for review.

As it stands however, I am sorry that I cannot be more positive.

Authors' correspondence

24th Jul 2019

I am writing to you after 6 months since your previous kind mail (that I attached below).

As you can imagine, along these months we have worked on this project collecting more data and reinforcing the manuscript. To be absolutely honest, soon after your decision we sent the manuscript out to EMBO Reports that rejected it mainly because of the lack of an evident molecular mechanism. Being more interested in the translational aspects of these studies, we decided to not focus on mechanisms but mainly on fixing major concerns and on the translational issue. Indeed, in the lab we are interested in identifying novel pathogenic mechanisms and investigating new therapeutic strategies for MECP2-related disorders.

We have thus implemented and ameliorated several previous figures therefore filling gaps identified by EMBO reports. Of relevance, we have added images showing a ciliary defect also in the *Mecp2* null cerebellum, we have expanded the number of RTT human fibroblasts showing the presence of a ciliogenesis defect and, as you requested, we have rescued this phenotype by stabilizing microtubules.

Since the RTT community has decided that preclinical studies have to be performed on females and males using two different *Mecp2* lines, we have decided to focus on these time consuming and complex studies in the next months.

I strongly believe that our data are quite relevant for the RTT community both for therapy and for novel pathogenic mechanisms. Indeed, we have also initiated a thorough characterization of the cerebellum development in *Mecp2* deficient mice.

Considering all the above, I am writing to you asking whether you can be so kind to let us know whether you think it will be worth for us to send the paper again to EMBO Molecular Medicine or you suggest us to consider a different journal.

I thank you in advance for your work and for considering this mail.

Editor's correspondence

25th Jul 2019

Thank you for your e-mail. I'm happy to read that you have added in vivo data and human samples experiments to your study! In order to be absolutely sure whether I would send the paper out for peer-review, I'd like to see the full set of new figures and legends at the very least. Could you please send these back to me at your earliest convenience and I will assess them and get back to you promptly.

Authors' correspondence

30th Jul 2019

Thank you for being so kind. Please find enclosed the definitive compressed figures and their legends. Further, for your convenience I have included the draft of the paper that however still requires a little bit of work.

Editor's correspondence

6th Aug 2019

Thank you for sending the figures and drafted article. I have now read it and indeed appreciate the in vivo work, adding more depth to the findings.

I am not entirely sure whether the study would provide sufficient translational insights but I would be happy in letting the referees assess it for us.

Therefore, please feel free to upload it on our website and mention this invitation during submission.

Thank you for the submission of your manuscript to EMBO Molecular Medicine. We have now heard back from the two referees whom we asked to evaluate your manuscript.

You will see that while referee 2 is more supportive of publication, referee 1 has questions that we feel are very pertinent to the study. Of particular interest, we would like you to focus on showing causality and disease relevance to your observations. In addition, better imaging, statistical analyses, and controls are also needed to make the paper stronger.

Given the balance of these evaluations, we feel that we can consider a revision of your manuscript if you can address the issues that have been raised. Please note that it is EMBO Molecular Medicine policy to allow only a single round of revision and that, as acceptance or rejection of the manuscript will depend on another round of review, your responses should be as complete as possible.

***** Reviewer's comments *****

Referee #1 (Remarks for Author):

The manuscript of Frasca et al. describes the investigation of the role of MeCP2 in ciliogenesis. The authors hypothesize that proper ciliary formation and maintenance requires functional Mecp2. Consistently, they found cilia alterations in cells lacking Mecp2, cells with Mecp2 KD and cells with Rett-associated mutations. In addition, the cilia phenotype could be rescued by expression of MeCP2 or treatment with the HDAC6 inhibitor tubacin. These novel observations might be of interest for the field of Rett syndrome.

The manuscript has some issues, though:

1-Not clear what figure 1 is attempting to show: there is positive signal for both, gamma and acetylated tubulin in all cells in the Mecp2 null panel and the reduction in length is not evident at all. An image more representative of the quantification data presented should be included.

2- What is the percentage of cells with silenced Mecp2 in the siRNA experiment?

3- In Fig 2, the choice of a green secondary antibody (to anti-AC3) in the background of GFP expression seems rather inconvenient.

4- In figure 3D, it seems that Mecp2 null MEFS show a very significant induction of Gli1 (the phrase "only weakly induced" does not represent the image).

5-In fig 5 A and B, P values should be reported for the post-hoc comparison of TUB vs VEH groups (instead of untreated), which seem to be significant as well.

6- The data from RT-PCR for MECP2 mRNA followed by quantitative sequencing (not shown) that confirm the data in cells with truncating and frameshift mutations, would be particularly informative to interpret the results for the missense mutations.

7- Rett is mostly a girls' disease, have the authors attempted to look at ciliogenesis in mouse Mecp2 +/- brains?

The actual relevance of the ciliary defect in the Rett phenotype is unknown, which reduces somehow the enthusiasm for this otherwise interesting paper. Manipulation of ciliogenesis in the mouse model (as proposed by the authors) followed by physiological and behavioral testing might be important to determine the role of the described defects in the disease. However, the investigation of cellular phenotypes (such as transcriptional patterns or electrophysiological parameters) could also suggest functional relevance to Rett syndrome.

The possibility of a secondary effect of cellular dysfunction on ciliogenesis or cilia maintenance (rather than a direct involvement of MeCP2 in ciliogenesis) is also not ruled out. Fibroblasts derived from patients other than Rett syndrome could serve as specificity control, to eliminate the possibility

that any genetic perturbation that affects cell function could result in a reduction in cilia length, similar to what is seen in the Rett missense mutants.

Referee #2 (Comments on Novelty/Model System for Author):

In this manuscript, the authors used MeCP2 knockout mouse. This is only one animal model of Rett syndrome.

Referee #2 (Remarks for Author):

Previously, Landsberger's group reported that MeCP2 localizes at the centrosome and is required for proper mitotic spindle organization (Baergo et al., JBC 2015). Now, Frasca et al. have shown that MeCP2 deficiency affects ciliogenesis and also the cilia-related Sonic Hedgehog pathway. In addition, the authors showed that microtubule instability participates to these phenotypes which can be rescued by HDAC6 inhibition.

All the experiments seem to be performed carefully, and the findings apparently contribute to understanding the new role of MeCp2 in Rett syndrome phenotypes.

I have one minor comment. Increased HDAC6 in MeCp2-deficient cells appears to be important for the defect of ciliogenesis. The authors need to discuss, at least, about the mechanism by which Mecp2 deficiency causes up-regulation of HDAC6. Some readers may speculate that a loss of MeCp2 at centrosome induces an increased HDAC6, while other authors may think that a loss of transcription repressor function of MeCP2 causes this. Since the authors appear to prefer the former possibility, I would recommend describe this clearly.

2nd Revision - authors' response

30th Jan 2020

Referee #1

The manuscript of Frasca et al. describes the investigation of the role of MeCP2 in ciliogenesis. The authors hypothesize that proper ciliary formation and maintenance requires functional Mecp2. Consistently, they found cilia alterations in cells lacking Mecp2, cells with Mecp2 KD and cells with Rett-associated mutations. In addition, the cilia phenotype could be rescued by expression of MeCP2 or treatment with the HDAC6 inhibitor tubacin. These novel observations might be of interest for the field of Rett syndrome.

The manuscript has some issues, though:

1-Not clear what figure 1 is attempting to show: there is positive signal for both, gamma and acetylated tubulin in all cells in the Mecp2 null panel and the reduction in length is not evident at all. An image more representative of the quantification data presented should be included.

2- What is the percentage of cells with silenced Mecp2 in the siRNA experiment?

We agreed with the reviewer that the original Fig.1 was not conveying the desired message. We have thus selected novel images for panels 1A, D, I and L, certainly improving the quality of this figure. We believe that the novel panels well represent the reported reduction in the number of ciliated cells and in the length of primary cilia. Furthermore, as correctly requested by the reviewer, through immunofluorescence we have determined the percentage of interfered cells in which MeCP2 is no more detectable. As reported in the novel panel 1F, almost 60% of cells appear well silenced, a result that fits quite well with the western blot shown in panel E.

3- In Fig 2, the choice of a green secondary antibody (to anti-AC3) in the background of GFP expression seems rather inconvenient.

The reviewer is absolutely correct. Actually, original data were acquired using a red secondary antibody to detect AC3, but by mistake we showed an old figure in which green was used. We

apologize for the mistake and thank the reviewer for noticing it. A novel Fig.2A has been generated.

4- In figure 3D, it seems that *Mecp2* null MEFS show a very significant induction of Gli1 (the phrase "only weakly induced" does not represent the image).

We have re-written the phrase according to the reviewer's suggestion and also modified panel 3D selecting a more representative image. The correction included in the manuscript is highlighted below:

Indeed, while a significant increase in Gli1 expression was measured both in WT MEFs and neurons (** $p < 0.001$, two-way ANOVA followed by Bonferroni post-hoc test), the activation of the Shh pathway did not lead to a significant Gli1 induction in *Mecp2* null cells. Accordingly, Gli1 levels resulted significantly reduced in SAG treated *Mecp2* null cells with respect to WT (* $p < 0.05$; ** $p < 0.01$, two-way ANOVA followed by Bonferroni post-hoc test).

5-In fig 5 A and B, P values should be reported for the post-hoc comparison of TUB vs VEH groups (instead of untreated), which seem to be significant as well.

As the reviewer suggested, we have performed the post-hoc comparison of TUB versus VEH groups and reported the significant results in the slightly modified Fig. 5A and B.

6- The data from RT-PCR for MECP2 mRNA followed by quantitative sequencing (not shown) that confirm the data in cells with truncating and frameshift mutations, would be particularly informative to interpret the results for the missense mutations.

In order to respond to the reviewer's comment, we performed deep sequencing of amplicons obtained from MeCP2 cDNAs. However, since the original cDNAs were exhausted, we had to obtain them from novel cultures. We thus discovered that fibroblasts carrying the missense mutation which were not anymore at the original early passage, have selected growth of cells expressing the wt allele. Of note, previous unpublished results revealed a concordant selection in a diverse RTT cell line, a result that might be explained by the proliferation defects characterizing MeCP2 deficient cells (Bergo et al. 2015).

Considering the impossibility to provide an adequate response to the reviewer and persuaded that the paper does not suffer from the lack of these two mutants, we have removed any data related to these cells. Anyhow, we find it relevant to observe that the inclusion in the manuscript of the novel Fig. EV1 confirms that missense mutations in the MBD of *Mecp2* do affect ciliogenesis.

7- Rett is mostly a girls' disease, have the authors attempted to look at ciliogenesis in mouse *Mecp2* +/- brains?

As requested by the reviewer, we have now included in the study a panel proving that the length of the primary cilium is significantly shorter also in *Mecp2*^{+/-} developing cerebella. Data are included in the novel panels D and F of figure 4. These novel data clearly prove that the defect is present in heterozygous females although, as expected, is reduced in magnitude.

The actual relevance of the ciliary defect in the Rett phenotype is unknown, which reduces somehow the enthusiasm for this otherwise interesting paper. Manipulation of ciliogenesis in the mouse model (as proposed by the authors) followed by physiological and behavioral testing might be important to determine the role of the described defects in the disease. However, the investigation of cellular phenotypes (such as transcriptional patterns or electrophysiological parameters) could also suggest functional relevance to Rett syndrome.

We thank the reviewer for the interest in our paper and for having prompted these assays, certainly improving the manuscript. In fact, we have now added the novel figure 6 showing that in concomitance with restored ciliogenesis, HDAC6 inhibition rescues the typical dendritic branching defects that characterize *Mecp2* null neurons. Further, we have included the TC-S 7010 inhibitor, which is characterized by a more specific HDAC6 inhibition at the primary cilium level. Our data highlight that TC-S 7010 is significantly more effective than tubacin in ameliorating synaptic phenotypes of *Mecp2* null neurons. In fact, TC-S 7010, but not tubacin, reverted the pre- and post-synaptic defects, together with the density of colocalized Synapsin1/2 and Shank2 puncta, therefore suggesting the recovery of mature synapses.

The possibility of a secondary effect of cellular dysfunction on ciliogenesis or cilia maintenance (rather than a direct involvement of MeCP2 in ciliogenesis) is also not ruled out. Fibroblasts derived from patients other than Rett syndrome could serve as specificity control, to eliminate the possibility that any genetic perturbation that affects cell function could result in a reduction in cilia length, similar to what is seen in the Rett missense mutants.

Our data indicate that HDAC6 inhibition can rescue ciliogenesis in *Mecp2* deficient cells, therefore indicating the involvement of a secondary effect. However, we have also proved that MeCP2 deficiency induces a primary cilium phenotype in any tested cell and that the expression of exogenous MeCP2 in null neurons can rescue the observed phenotypes. These data directly link the expression of the methyl binding protein with ciliogenesis.

MeCP2 is a multifunctional protein and its deficiency is expected to affect several molecular pathways and the related cellular features. Because of that Rett research is trying to identify direct or indirect deregulated mechanisms that might be pharmaceutical targeted. This was the main aim of our study. Considering all above, the high heterogeneity of human fibroblasts (differing for gender, age and genotype), the difficulty in selecting an appropriate disorder, we found not convenient to analyze fibroblasts modeling other diseases. If useful, we can mention that we had previously analyzed ciliogenesis in *Cdkl5* deficient cells, a validated model for studying the CDKL5 deficiency disorder, not revealing the same phenotypes.

Referee #2

(Comments on Novelty/Model System for Author):

In this manuscript, the authors used MeCP2 knockout mouse. This is only one animal model of Rett syndrome.

Although the reviewer did not ask for it, we found it useful to assess whether a defect in ciliogenesis could also be observed in a second mouse model of *Mecp2*. To this purpose we used MEF cells obtained from the *Mecp2*^{Y120D/Y} mouse model that we recently generated (Gandaglia et al., 2018). Our data, which are now included in the novel EV1 figure, confirmed the presence of a ciliary defect.

I have one minor comment. Increased HDAC6 in MeCP2-deficient cells appears to be important for the defect of ciliogenesis. The authors need to discuss, at least, about the mechanism by which MeCP2 deficiency causes up-regulation of HDAC6. Some readers may speculate that a loss of MeCP2 at centrosome induces an increased HDAC6, while other authors may think that a loss of transcription repressor function of MeCP2 causes this. Since the authors appear to prefer the former possibility, I would recommend describe this clearly.

We thank the reviewer for the positive comments. As requested, we made clearer that HDAC6 up-regulation is a post-transcriptional mechanism caused by MeCP2 deficiency. We also mentioned the fact that this might be caused by defects in miRNA expression; in fact, it is well recognized an involvement of MeCP2 in non-coding RNA transcription and maturation. For convenience, we are including the related novel version of the discussion.

Importantly, the presence of primary cilium defects was confirmed in heterozygous RTT patients' fibroblasts characterized by premature stop codons or a frameshift mutation occurring in the TRD and in *Mecp2*^{Y120D/Y} MEFs, harbouring a *Mecp2* missense mutation that mainly affects the capacity of the protein to tightly associate with DNA and chromatin (D'Annessa et al., 2018; Gandaglia et al., 2018). We thus hypothesize that the transcriptional functions of MeCP2 are involved in the observed ciliary defects, although an involvement of its centrosomal functions cannot be excluded. Indeed, we have recently shown that in the absence of MeCP2, microtubule polymerization from centrosomes is impaired (Bergo et al., 2015). In good accordance, a reduction in microtubule stability associated with decreased levels of α -tubulin acetylation, caused by increased levels of HDAC6, was reported (Delépine et al., 2016; Gold et al., 2015; Landucci et al., 2018; Xu, Alan, & Pozzo-Miller, 2014). Real time RT-PCR suggested that a post-transcriptional mechanism is involved in the observed HDAC6 upregulation (Delépine et al., 2016) and our unpublished data confirm that *Hdac6* mRNA levels are not perturbed in the *Mecp2* null brain. Of note, although the molecular

mechanisms causing this aberrant HDAC6 activity remain unknown, it has been already reported that miRNA dysregulation can lead to HDAC6 overexpression (Mansini et al., 2018) and *Mecp2* deficiency

3rd Editorial Decision

9th Mar 2020

Thank you for the submission of your revised manuscript to EMBO Molecular Medicine. We have now received the enclosed reports from the referees that were asked to re-assess it. As you will see the reviewers are now globally supportive and I am pleased to inform you that we will be able to accept your manuscript pending the following final amendments:

1) Please address the minor comments of referee 1.

Please provide a point-by-point letter INCLUDING my comments as well as the reviewer's reports and your detailed responses to their comments (as Word file).

***** Reviewer's comments *****

Referee #1 (Remarks for Author):

This re-submission is an enhanced version of the manuscript since the authors were responsive to reviewers' suggestions. Some concerns remain unsolved, that when clarified would improve the paper.

-the size of the cilia should be related to the soma size, specially in the case of MeCP2 mutant cells, in which soma size alterations (reduction) were reported. It is important to determine whether there is a propositional decrease in cilium length or is just a reflection of cell size reduction.

- The interesting observation of reduced cilium length in heterozygous *Mecp2* mice will greatly benefit from a description of the phenotype in terms of its penetrance. If most cells have reduced cilia length, a non-cell autonomous mechanism needs to be invoked due to the functional mosaicism caused by X chromosome inactivation, whereas approximately 50% have smaller cilia and 50% have regular sized cilia, the effect is likely cell autonomous.

-Not clear the connection between in vitro dendrite complexity and ciliary alterations. The authors need to clarify. Also, they should consider that the activity of TUB and TC-S on number of synapses and dendrites is independent of cilia formation.

-There are grammatical errors such as "participates to...", including in the abstract section that need to be corrected.

Referee #2 (Comments on Novelty/Model System for Author):

The authors appropriately used MeCP2 mutant mice, which have been widely used as the model of Rett syndrome.

Referee #2 (Remarks for Author):

In this revised manuscript, the authors have adequately discussed that increased HDAC6 in MeCP2-deficient cells is important for the defect of ciliogenesis. Thus, the authors have appropriately replied to my comments.

3rd Revision - authors' response

19th Mar 2020

The size of the cilia should be related to the soma size, especially in the case of MeCP2 mutant cells, in which soma size alterations (reduction) were reported. It is important to determine whether there is a propositional decrease in cilium length or is just a reflection of cell size reduction.

This is an interesting observation and we thank the reviewer for raising the point. To address this issue, we have measured the soma area of neurons, confirming its well-known defect. On the contrary, and as expected, patients' fibroblasts did not show any reduction in the soma size. By counting ~20 cells for genotype, we have then measured in neurons and human fibroblasts the soma area and the length of its primary cilium, reporting no correlation between the two variables. Data have been reported in the Appendix figure S1 and commented in the discussion.

The interesting observation of reduced cilium length in heterozygous *Mecp2* mice will greatly benefit from a description of the phenotype in terms of its penetrance. If most cells have reduced cilia length, a non-cell autonomous mechanism needs to be invoked due to the functional mosaicism caused by X chromosome inactivation, whereas approximately 50% have smaller cilia and 50% have regular sized cilia, the effect is likely cell autonomous.

We agreed with the reviewer that this is an important issue. We have thus calculated in our heterozygous brain samples the % of cells expressing *Mecp2*, obtaining an average of ~40%. This observation, together with the fact that our data of cilium length in the het brains (**Fig.4F**) do not have a bimodal but a normal distribution (verified by D'Agostino-Pearson normally test) prompted us to suggest in the Discussion the occurrence of a non-cell autonomous mechanism.

Not clear the connection between in vitro dendrite complexity and ciliary alterations. The authors need to clarify. Also, they should consider that the activity of TUB and TC-S on number of synapses and dendrites is independent of cilia formation.

The connection between dendritogenesis and ciliary alterations is now clarified in Results mainly referring to the Guadiana et al. paper (2013). The same point has been extended in the Discussion, also including a sentence clarifying that unknown mechanisms might be involved in the positive effects exerted by the two drugs.

There are grammatical errors such as "participates to...", including in the abstract section that need to be corrected.

We have made any effort to remove grammatical errors; we apologize for them.

4th Editorial Decision

26th Mar 2020

We are pleased to inform you that your manuscript is accepted for publication and will soon being sent to our publisher to be included in the next available issue of EMBO Molecular Medicine.

Corresponding Author Name: NICOLETTA LANDSBERGER

Journal Submitted to: EMBO MOLECULAR MEDICINE

Manuscript Number: EMM-2019-10270